# USP7 deubiquitinase stabilizes FAN1 to support DNA crosslink repair and suppress CAG repeat expansion

Giulio Collotta [1], Marco Gatti [1], Irina-Maria Ungureanu[2], Vanessa van Ackeren [1], Emilie Rannou[3], Francesca Vivalda [1], Diego Gomez Vieito[1], Keri M. Fishwick[1], Christine von Aesch[1], Antonio Porro[1], Kyra Ungerleider[2], Ailin Heidari [4,5,6], Raphaël Guérois [7], Rachel J. Harding [4,5,6], Sylvain Bischof [3], Gabriel Balmus [2,8] ✉ & Alessandro A. Sartori [1] ✉

Human FAN1 is a structure-specific endonuclease implicated in the repair of DNA interstrand crosslinks (ICLs) and the excision of extrahelical CAG repeats–whose pathological expansion underlies Huntington's disease (HD), a progressive and currently incurable neurodegenerative disorder. However, mechanisms of post-translational regulation of FAN1 are still largely unknown. Here, we identify the ubiquitin-specific protease 7 (USP7) as an interactor of FAN1. USP7 stabilizes FAN1 protein levels in a deubiquitination-dependent manner, preventing FAN1 from proteasomal degradation. Consequently, we demonstrate that USP7 depletion leads to reduced chromatin association of FAN1 and increased cellular hypersensitivity following ICL damage. Moreover, loss of USP7 accelerates CAG repeat expansion in an RPE-1 cell model stably expressing mutant huntingtin (*mHTT*) exon 1 containing 129 CAG repeats (RPE-1[HTT-CAG129]). Collectively, our findings uncover a link between USP7 and FAN1 in mechanisms that preserve genome stability and influence repeat instability.

FAN1 (FANCD2 and FANCI-associated nuclease 1) was originally identified by four independent research groups as a DNA repair enzyme involved in the processing of DNA interstrand crosslinks (ICLs)[1–4]. FAN1 possesses dual catalytic activity, functioning both as a 5′ flap endonuclease and a 5′ − 3′ exonuclease. This enables FAN1 to unhook ICLs, presumably through a series of precise incisions spaced approximately three nucleotides apart[5–7]. Beyond its role in ICL repair, FAN1 also interacts with mismatch repair (MMR) proteins of the MutL family, including MLH1, PMS2, and MLH3, forming MutLα and MutLγ nuclease complexes[2,3,8]. Notably, MLH1 binding was proposed to restrict FAN1's nuclease activity to specific DNA substrates, as cells expressing FAN1 mutants defective in MLH1 interaction displayed increased sensitivity to ICL-inducing agents[9,10]. More recently, genome-wide association studies (GWAS) have shown that FAN1 variants are the strongest genetic modifiers in Huntington's disease (HD), a hereditary and fatal neurodegenerative disorder[11]. Here, FAN1 is thought to compete with the MMR system and suppress somatic expansion of CAG tracts in exon 1 of mutant *Huntingtin* (*mHTT*)[12–16]. Notably, elevated FAN1 expression have been associated with delayed age-of-onset in HD patients, correlating with a slower rate of CAG repeat expansion[17].

[1]Institute of Molecular Cancer Research, University of Zurich, Zurich, Switzerland. [2]UK Dementia Research Institute, Department of Clinical Neurosciences, University of Cambridge, Cambridge Biomedical Campus, Cambridge, UK. [3]Department of Plant and Microbial Biology, University of Zurich, Zurich, Switzerland. [4]Structural Genomics Consortium, University of Toronto, Toronto, ON, Canada. [5]Department of Pharmacology & Toxicology, University of Toronto, Toronto, ON, Canada. [6]Leslie Dan Faculty of Pharmacy, University of Toronto, Toronto, ON, Canada. [7]Université Paris-Saclay, CEA, CNRS, Institute for Integrative Biology of the Cell (I2BC), Gif-sur-Yvette, France. [8]Department of Molecular Neuroscience, Transylvanian Institute of Neuroscience, Cluj-Napoca, Romania. ✉e-mail: gb318@cam.ac.uk; sartori@imcr.uzh.ch

Cellular FAN1 protein levels are relatively low and, typical for enzymes capable of cleaving DNA, its activity expected to be tightly regulated through post-translational modifications, including proteolysis via the ubiquitin-proteasome pathway. The ubiquitination machinery has emerged as an important player in maintaining genome stability, orchestrating essential DNA damage response (DDR) processes, including multiple DNA repair pathways[18]. Despite this, very little is known about the specific factors that govern FAN1 ubiquitination, aside one report describing APC/C-Cdh1−dependent FAN1 degradation when cells exit mitosis[19]. Protein ubiquitination is a highly dynamic and reversible process, primarily regulated by selective E3 ubiquitin ligases and deubiquitinases (DUBs), which add or remove ubiquitin chains from target proteins, respectively[20]. The human genome encodes around 100 DUBs, categorized into six distinct families. Among these, the ubiquitin-specific protease (USP) family is the largest, with many members being associated with all hallmarks of cancer, most notably genomic instability and mutation[21].

In this study, using mass spectrometry-based proteomics, we identified the DUB USP7 as a regulator of FAN1. We show that USP7 interacts with FAN1 through its N-terminal TRAF-like and C-terminal ubiquitin-like (UBL) domains. USP7 deubiquitinates FAN1, thereby preventing its proteasomal degradation. Moreover, we demonstrate that USP7 is required for the assembly of FAN1 at damaged chromatin following ICL damage. Loss of USP7 results in elevated apoptotic signalling and hypersensitivity to mitomycin C (MMC). Lastly, we provide evidence that USP7 is necessary for maintaining CAG repeat stability. Collectively, these findings uncover a mechanism by which USP7 stabilizes FAN1, offering insights into the post-translational regulation of DNA repair proteins in human cells.

## Results

### Identification of USP7 as an interaction partner of FAN1

To gain insights into the regulation of human FAN1, we interrogated the FAN1 interactome using affinity purification coupled to mass spectrometry (AP-MS). To this end, we established a HEK293 cell line, in which endogenous FAN1 was replaced by an inducible, eGFP-tagged variant, using a stably-integrated doxycycline (Dox)-inducible vector containing two cassettes: one expressing GFP-FAN1 wild-type (wt), and the other an shRNA targeting endogenous *FAN1* mRNA[22,23]. Using this cell line, we performed GFP-Trap co-immunoprecipitation (co-IP) from nuclei-enriched lysates followed by liquid chromatography-tandem mass spectrometry (LC-MS/MS). This analysis recovered known FAN1 interactors, including the MutLα complex (MLH1-PMS2)[8], the FANCD2/FANCI complex[1–4], and PCNA[23] (Fig. 1A and Supplementary Data 1). In addition to these established interactors, we also discovered several putative FAN1 regulators, including three USP-type deubiquitinases (USP7, USP9X and USP48), all previously implicated in the DNA damage response[24–27] (Fig. 1A and Supplementary Data 1). USP7 and USP9X have also been detected, but not validated, in an earlier FAN1 AP-MS study[4].

To validate these proteomics data, we performed GFP-Trap pulldowns in HEK293^eGFP-FAN1 cells and found that endogenous USP7, USP9X, and USP48 co-precipitated with FAN1 (Fig. 1B). To determine whether these DUBs influence FAN1 stability, we individually depleted USP7, USP9X and USP48 from U2OS^eGFP-FAN1 cells. As a control, we included USP11, a DUB not enriched in our AP-MS data set but previously implicated in DNA repair[28,29]. Western blot analysis revealed that USP7 silencing caused a pronounced reduction in FAN1 protein levels, whereas USP9X depletion produced only a mild decrease, and depletion of the remaining DUBs did not affect FAN1 stability (Fig. 1C). Reciprocal pulldown experiments further demonstrated that the USP7-FAN1 interaction occurs independently of USP9X (Figure S1A), despite prior reports that USP7 and USP9X cooperate in the deubiquitination of certain substrates[27].

Given its robust impact on FAN1 protein levels, USP7 was prioritized for further functional investigation. Co-IP of FLAG-USP7 from

HEK293 cells confirmed its association with endogenous FAN1 (Fig. 1D), and reciprocal co-IP using an anti-FAN1 antibody in nuclear extracts from HeLa cells corroborated this interaction under physiological conditions (Fig. 1E). To address whether the USP7-FAN1 interaction is modulated by DNA damage, we performed in situ proximity ligation assays (PLA) in HeLa parental and HeLa FAN1 knock-out (KO) cells treated with mitomycin C (MMC). PLA signals significantly increased in numbers upon MMC treatment, suggesting that the FAN1-USP7 association is enhanced following ICL damage (Fig. 1F). To map the FAN1 region required for USP7 binding, we performed GFP-Trap pulldowns using FAN1 truncation constructs in HEK293 cells. USP7 associated with the FAN1 N-terminal domain (aa 1-372, NTD), a largely unstructured region of FAN1, while deletion of this region (aa 373-1017, ΔNTD) abolished USP7 binding (Fig. 1G). As the FAN1-NTD contains two MLH1-binding motifs[9], we asked whether USP7-FAN1 interactions depend on MLH1. A synthetic, 60mer FAN1 peptide covering the entire MLH1-binding region (aa 118-177) blocked FAN1- MLH1, but not FAN1-USP7 interaction, indicating that USP7 binds to FAN1 in an MLH1-independent manner (Figure S1B). Consistently, FAN1-USP7 interaction persisted in MLH1-deficient HEK293T cells (Figure S1C). Moreover, treatment with the ubiquitin-activating enzyme (UAE) inhibitor TAK-243[30], causing depletion of cellular ubiquitin conjugates, did not impact FAN1-USP7 binding, indicating this interaction is largely independent of FAN1 ubiquitination (Figure S1C). Finally, GST-FAN1 pulldown experiments using purified recombinant proteins confirmed a physical interaction between recombinant USP7 and the FAN1-NTD (Fig. 1H). Collectively, these findings identify USP7 as a FAN1-interacting protein and suggest that USP7 stabilizes FAN1 through direct binding to its N-terminal domain. This interaction is enhanced upon DNA damage and may represent a regulatory axis in the cellular response to genotoxic stress.

### FAN1 associates with both the TRAF-like and UBL domains of USP7

Substrate recognition by USP7, including RNF169, p53 and DNMT1 among many others, is typically mediated through two evolutionarily conserved regions: the N-terminal TRAF-like domain and the tandem UBL1-2 domains at the C-terminus[31–34]. While most substrates preferentially bind one of these domains, a subset−including DNA polymerase iota (Pol ι) and PAF15−has been shown to engage both[35,36]. To assess which USP7 domain interacts with FAN1, we transfected FLAG-tagged full-length (FL) USP7 constructs bearing point mutations in either the TRAF (D164A/W165A) or UBL2 (D762R/D764R) domain[37] into HEK293^eGFP-FAN1 or HEK293T cells and performed FLAG-IPs (Fig. 2A and S2A). Compared to well-characterized USP7 substrates, FAN1 binding appeared weaker and less sharply defined (Fig. 2A and S2A). Nonetheless, the interaction was markedly reduced upon mutation of the TRAF-like domain, indicating that this domain is a principal site of FAN1 recognition, as seen for p53 (Fig. 2A and S2A). Structural studies of USP7 binding partners have revealed a conserved P/A/ExxS linear sequence motif as a core recognition element for the TRAF-like domain[32,38,39]. Thus, we conducted in silico phylogenetic analysis of mammalian FAN1 homologs and identified four candidate regions matching this motif in the FAN1-NTD (Figure S2B). To functionally assess these, we synthesized biotinylated 12mer FAN1 peptides harbouring individual P/A/ExxS motifs and tested their binding to recombinant USP7 using streptavidin pulldown assays (Fig. 2B). Remarkably, only the FAN1 peptide containing the 181-PQSS-184 TRAF-like domain binding motif interacted with USP7 (Fig. 2B). Moreover, mutation of the key serine residue 184 to alanine (S184A) abrogated USP7 binding (Fig. 2C). Consistently, AlphaFold 3 (AF3)[40] structural predictions suggest that FAN1-S184 directly contacts the USP7 substrate-binding groove, including residue D164, a critical component of the TRAF-binding interface (Figures S2C and S2D). To further define the molecular determinant mediating FAN1 recognition by

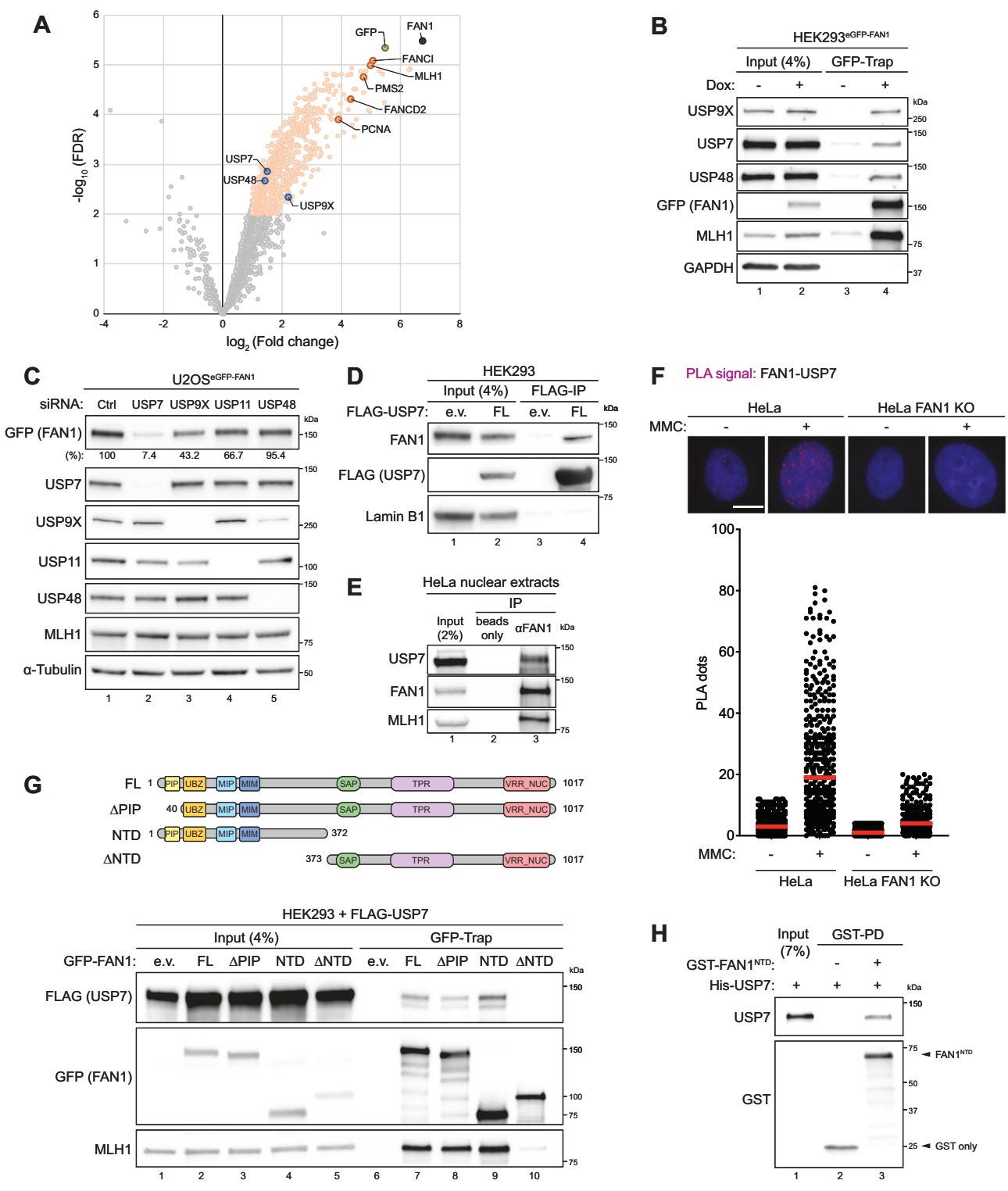

USP7, we investigated whether the 181-PQSS-184 sequence in FAN1 functions as a canonical PxxS TRAF-binding motif. To this end, we performed fluorescence polarisation (FP) assays using synthetic 8mer peptides encompassing this region. As shown in Fig. 2D, the wild-type peptide bound efficiently to the USP7 TRAF domain, whereas substitution of the serine residue (S184A) markedly impaired binding. In contrast, the P181A mutation did not abolish the interaction, consistent with the ability of an AxxS sequence to conform to the TRAF consensus motif (Fig. 2D). Together, these results validate 181-PQSS-184 as a canonical TRAF-binding element required for robust FAN1

engagement by USP7. However, full-length GFP-FAN1 carrying the S184A mutation retained the ability to co-precipitate with endogenous USP7, indicating the presence of additional or alternative binding interfaces (Figure S2E). Consistent with this notion, a FLAG–USP7 variant bearing TRAF-domain mutations (D164A/W165A) still weakly co-immunoprecipitated GFP-FAN1 as well as endogenous FAN1 (Fig. 2A and S2A). These observations prompted us to examine whether other USP7 domains are involved in mediating FAN1 binding.

To that end, we expressed a series of USP7 truncation mutants in HEK293^eGFP-FAN1 cells and performed FLAG co-IPs (Fig. 2E). In contrast to

**Fig. 1 | USP7 interacts with the N-terminus of FAN1. A** Volcano plot showing all GFP-FAN1-interacting proteins identified by immunoprecipitation with GFP-Trap beads followed by mass spectrometry in HEK293^eGFP-FAN1 cells (n = 3). Significantly enriched proteins (log_2 (Fold Change) ≥ 1 and -log_10 (FDR) ≥ 2) are displayed as light orange dots. FAN1, GFP, known FAN1 interactors and DUBs are shown as black, green, dark orange and blue dots respectively. **B** HEK293^eGFP-FAN1 cells were induced with doxycycline (Dox, 100 ng/ml). 48 h later, cells were lysed and whole-cell extracts were subjected to GFP-Trap resin. Inputs and recovered protein complexes were analysed by immunoblotting. **C** Doxycycline-inducible U2OS^eGFP-FAN1 cells were transfected with either non-targeting (Ctrl) or indicated siRNA oligos. 24 h later, GFP-FAN1 expression was induced with Dox (100 ng/ml) for 24 h and whole-cell lysates were analysed by immunoblotting. Relative GFP-FAN1 protein levels were determined by quantification of GFP-FAN1 band intensity (normalized to α-Tubulin) with the ImageJ software. **D** HEK293 cells were transfected with either empty vector (e.v.) or the FLAG-USP7-wt expression constructs. 48 h after transfection, cells were lysed and whole-cell extracts were subjected to IP using anti-FLAG M2 affinity resin. Inputs and recovered protein complexes were analysed by immunoblotting. **E** HeLa nuclear extracts were subjected to immunoprecipitation using anti-FAN1 antibody. Inputs and recovered protein complexes were analysed by immunoblotting. **F** PLA was used to evaluate FAN1-USP7 association in HeLa

parental and FAN1 KO cells, mock-treated or treated with MMC (150 ng/ml) for 24 h. Dot plot shows the number of PLA foci and the median (red line) from one biological replicate, analysing at least 260 cells for each condition. Representative images are shown. Scale bar: 10 µm. This experiment was repeated three times. **G** Schematic representation of the human FAN1 protein indicating truncation mutants. PIP proliferating cell nuclear antigen-interacting peptide box (yellow), UBZ ubiquitin-binding zinc-finger domain (orange), MIP MLH1-interacting protein box (cyan), MIM MLH1-interacting motif (blue), SAP SAF-A/B, Acinus and PIAS domain (green), TPR tetratricopeptide repeat domain (violet), VRR_NUC virus-type replication-repair nuclease domain (red), FL full-length, NTD N-terminus, ΔNTD deleted NTD. HEK293 cells were co-transfected with FLAG-USP7 and either e.v. or the indicated GFP-FAN1 expression constructs. 48 h after transfection, cells were lysed and whole-cell extracts were subjected to GFP-Trap resin. Inputs and recovered protein complexes were analysed by immunoblotting. **H** Soluble extracts expressing either GST or GST-FAN1-NTD (aa 1-372) were immobilized on Glutathione Sepharose 4B affinity resin and incubated with purified recombinant His-USP7. Inputs and recovered protein complexes were analysed by immunoblotting. Arrows indicate the corresponding GST-tagged protein. The experiments in (**B**–**E**, **G**, **H**) were repeated three independent times.

the domain-specific interactions observed for p53 (TRAF-like) and RNF169 (UBL1-2), GFP-FAN1 co-immunoprecipitated with all USP7 truncation mutants, albeit with reduced efficiency compared to full-length USP7 (Fig. 2E). These findings suggest a cooperative contribution of both the TRAF-like and UBL-domains to FAN1 binding. This dual engagement was confirmed using in vitro streptavidin pull-down experiments employing recombinant biotin-tagged USP7 TRAF-like or UBL domains, along with MBP-His-tagged FAN1 (Figure S3A). Accordingly, we searched for potential UBL1-2 domain binding sites within the FAN1-NTD that match the consensus KxxxK sequence motif, which has been previously identified in USP7 substrates such as RNF169 and DNMT1[31,41]. The Eukaryotic Linear Motif (ELM) prediction tool revealed five such motifs[42]. Two were excluded due to their location upstream of the FAN1 PIP-box region, which we had previously shown to be dispensable for USP7 interaction (Fig. 1G). Streptavidin-pull-down assays using biotinylated FAN1-derived 13mer peptides encompassing the remaining putative UBL-binding consensus motifs revealed that only the peptide containing 158-KLSRK-162 sequence exhibited detectable USP7 binding (Figure S3B). However, mutating the two lysine residues to alanine (K158A/K162A) did not abolish USP7 association (Figure S3C), indicating that while 158-KLSRK-162 is unlikely to constitute a 'bona fide' UBL-binding motif, this region in FAN1 still contributes to USP7 interaction. To validate these findings, we performed GFP-trap pulldown assays using HEK293 cells expressing FAN1 constructs of varying N-terminal lengths. We found that the FAN1^1-120 fragment did not interact with USP7, whereas FAN1^1-165 was the smallest fragment capable of binding to USP7, consistent with the inclusion of the 158-KLSRK-162 region (Figure S3D). USP7 binding was further enhanced in the FAN1^1-190 fragment, which also includes the 181-PQSS-184 TRAF-binding motif (Figure S3D). Deletion of a 50-aa region (residues 157-207) from the FAN1^1-290 fragment, encompassing both proposed USP7-binding sites, substantially disrupted the FAN1-USP7 interaction (Figure S3D). In contrast, targeted deletion of the same 50-aa region from full-length FAN1 did not impair USP7 binding (Figure S3E), suggesting that USP7 can engage alternative proximal TRAF- or UBL-binding motifs in the absence of canonical sites, potentially acting as a safeguard to ensure robust regulation of FAN1. Notably, PAF15 harbours two P/AxxS and a single KxxxK motif, and only a triple mutant abolished USP7 binding[36]. Together, our data demonstrate that FAN1 engages both the TRAF-like and UBL domains of USP7 through at least one distinct linear motif within its N-terminal region. This bipartite binding mode likely reflects a flexible and robust substrate recognition mechanism, ensuring dynamic regulation of FAN1 during DNA repair.

## USP7 regulates FAN1 stability through its deubiquitinase activity

Having established USP7 as a FAN1 interactor, we next sought to dissect its functional impact on FAN1 protein stability. Consistent with our initial observation that USP7 depletion reduced FAN1 levels in U2OS^eGFP-FAN1 cells (Fig. 1C), we investigated this relationship in greater detail using both genetic and pharmacological approaches. We transfected non-transformed, hTERT immortalized retinal pigment epithelial (RPE-1) cells, either parental or p53 KO, with two independent siRNA oligos targeting the coding sequence of USP7. In RPE-1 parental p53-proficient cells, USP7 knock-down led to increased p53 protein levels, presumably due to MDM2 degradation[43], and was accompanied by a pronounced G1 cell-cycle arrest (Fig. 3A and S4A). In contrast, p53 KO RPE-1 cells exhibited normal proliferation following USP7 silencing, confirming p53-dependence of the observed cell-cycle effects (Fig. 3A and S4A). Notably, FAN1 protein levels were strongly reduced in both cellular backgrounds, paralleling the behaviour of DNMT1, a known USP7 substrate (Fig. 3A)[33]. To further confirm USP7's role in FAN1 regulation, we asserted the impact of USP7 inhibition on ectopically expressed FAN1. Knock-down or catalytic inhibition using two non-covalent small molecule inhibitors for USP7[44,45], led to reduced expression of exogenous FAN1 in Dox-inducible U2OS^eGFP-FAN1 and HeLa^eGFP-FAN1 cancer cells (Fig. 3B, C and S4B). Importantly, FAN1 levels were restored in USP7-depleted U2OS^eGFP-FAN1 and RPE-1 cells upon treatment with the proteasome inhibitor MG132, indicating that USP7 protects FAN1 from proteasome-mediated degradation (Fig. 3D and S4C). To confirm these results, we examined the FAN1 protein half-life in U2OS^eGFP-FAN1 cells in the presence of the protein synthesis inhibitor cycloheximide (CHX). FAN1 exhibited a reduced protein half-life upon CHX treatment and this effect was exacerbated by USP7 knock-down (Figure S4D), suggesting that USP7 directly modulates FAN1 protein stability. To determine if USP7's catalytic activity is required for FAN1 stabilisation, we overexpressed USP7-wt or a catalytic-dead mutant (C223S) in HEK293^eGFP-FAN1 cells. Only USP7-wt increased GFP-FAN1 levels, while the C223S mutant had no effect (Fig. 3E). Consistently, overexpression of USP7-wt reduced FAN1 polyubiquitination in cells co-transfected with HA-tagged ubiquitin, whereas USP7-C223S failed to do so (Fig. 3F). To investigate the nature of FAN1 polyubiquitination chains, we employed an HA-ubiquitin mutant retaining only lysine 48 (K48) as acceptor site, with all other lysines mutated to arginine. USP7-wt supressed K48-linked polyubiquitination of FAN1 whereas the C223S mutant did not, implicating USP7 in the removal of K48-linked chains that signal proteasomal degradation (Fig. 3G). Furthermore, when endogenous ubiquitin was the only source for ubiquitination, overexpression of the catalytically-inactive USP7-C223S

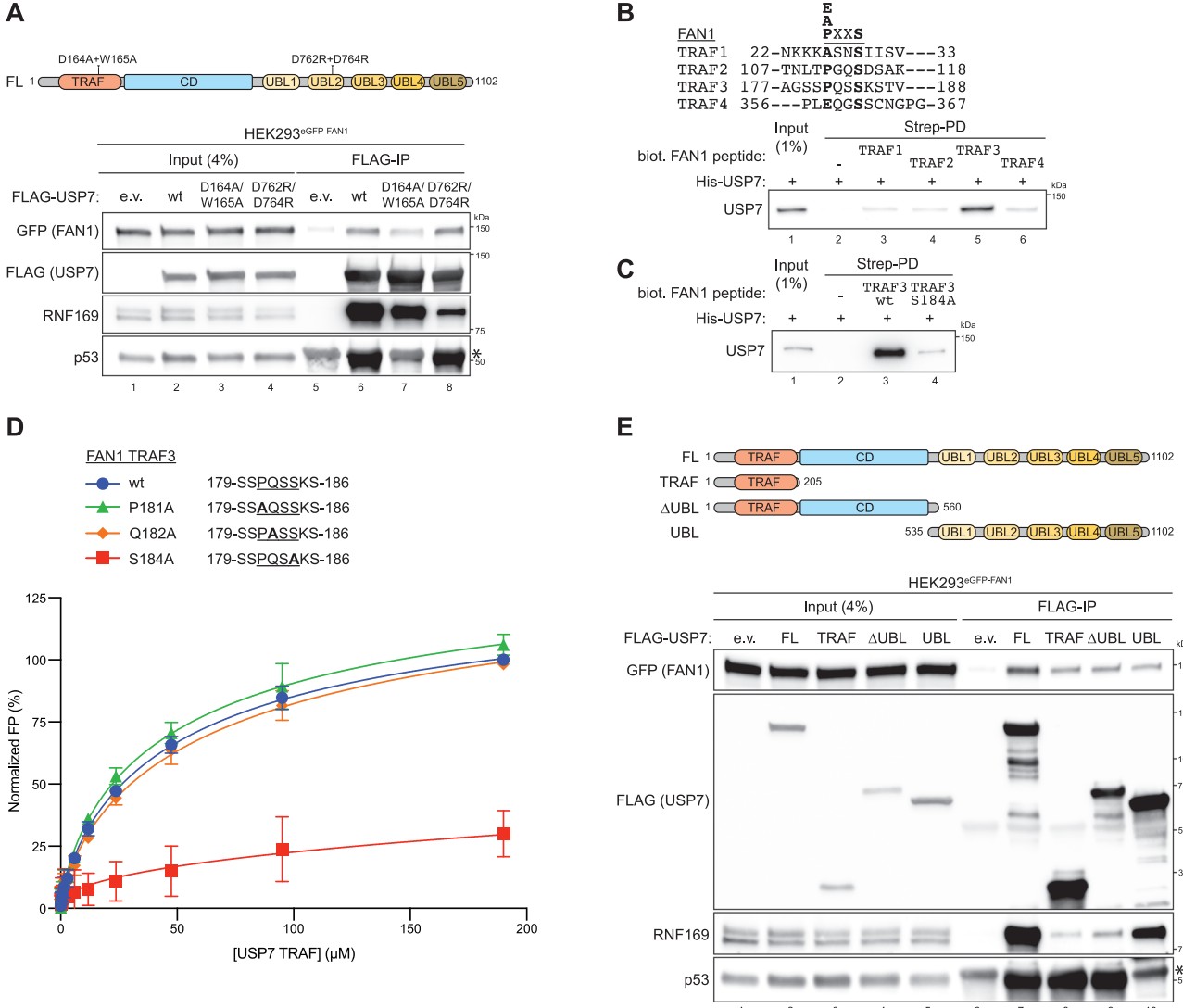

**Fig. 2 | FAN1 associates with the TRAF-like and UBL domains of USP7. A** Top: schematic representation of the human USP7 protein indicating point mutants. TRAF tumour necrosis factor receptor associated factor (orange), CD catalytic domain (cyan), UBL ubiquitin-like (yellow to brown). Bottom: HEK293$^{eGFP\text{-}FAN1}$ cells were transfected with either e.v. or the indicated FLAG-USP7 expression constructs. 24 h later, GFP-FAN1 expression was induced with Dox (100 ng/ml). 48 h post induction, cells were lysed and whole-cell extracts were subjected to IP using anti-FLAG M2 affinity resin. Inputs and recovered protein complexes were analysed by immunoblotting. Asterisk indicates IgG heavy chain from anti-FLAG M2 affinity resin. **B** Sequences of the biotinylated 12-mer FAN1-derived TRAF peptides. The corresponding amino acids are indicated. Key residues are highlighted in bold. **B, C** Biotinylated FAN1 TRAF peptides (10 μg) were immobilized on Strep-Tactin®XT-4Flow® resin and incubated with recombinant purified His-USP7. Inputs and recovered His-USP7 were analysed by immunoblotting. **D** Fluorescence polarization (FP) binding curves of FITC-labelled FAN1-derived 8mer peptides: wild-type (wt; black circle), P181A (green triangle), Q182A (orange diamond), S184A (red

square). The corresponding amino acids are indicated. Key residues are highlighted in bold. FP was measured as a function of USP7-TRAF (aa 2–207) concentration. FP values were normalized by expressing the millipolarization (mP) of each sample relative to the corresponding wild-type peptide. Background signal was corrected by subtracting baseline FP measured in the absence of USP7-TRAF. Data are presented as the means ± SD of three biological replicates ($n = 3$). **E** Top: schematic representation of the human USP7 protein indicating truncation mutants. TRAF tumour necrosis factor receptor associated factor, CD catalytic domain, UBL ubiquitin-like. HEK293$^{eGFP\text{-}FAN1}$ cells were transfected with either e.v. or the indicated FLAG-USP7 expression constructs. 24 h later, GFP-FAN1 expression was induced with Dox (100 ng/ml). 48 h post induction, cells were lysed and whole-cell extracts were subjected to IP using anti-FLAG M2 affinity resin. Inputs and recovered protein complexes were analysed by immunoblotting. Asterisk indicates IgG heavy chain from anti-FLAG M2 affinity resin. The experiments in (**A–C**, **E**) were repeated three independent times.

mutant led to an accumulation of polyubiquitinated FAN1 species, consistent with dominant-negative effect under these conditions (Figure S4E). Finally, we demonstrated that purified recombinant USP7 deubiquitinates immune-purified polyubiquitinated GFP-FAN1 in vitro (Fig. 3H). Collectively, our results establish USP7 as a bona fide DUB for FAN1. By removing K48-linked polyubiquitination chains, USP7 prevents proteasomal degradation of FAN1 via its enzymatic activity thereby maintaining FAN1 at steady state levels in cells.

## USP7 facilitates FAN1 localization and repair activity at ICLs

We earlier showed that MMC-induced DNA damage enhances the interaction between FAN1 and USP7, suggesting that USP7 may primarily function to stabilize or regulate FAN1 activity at sites of ICLs (Fig. 1F). To substantiate this possibility using an alternative readout, we scored chromatin-associated FAN1 foci in U2OS$^{eGFP\text{-}FAN1}$ cells by using quantitative image-based cytometry (QIBC), a method previously shown to effectively monitor the re-localization of FAN1 in

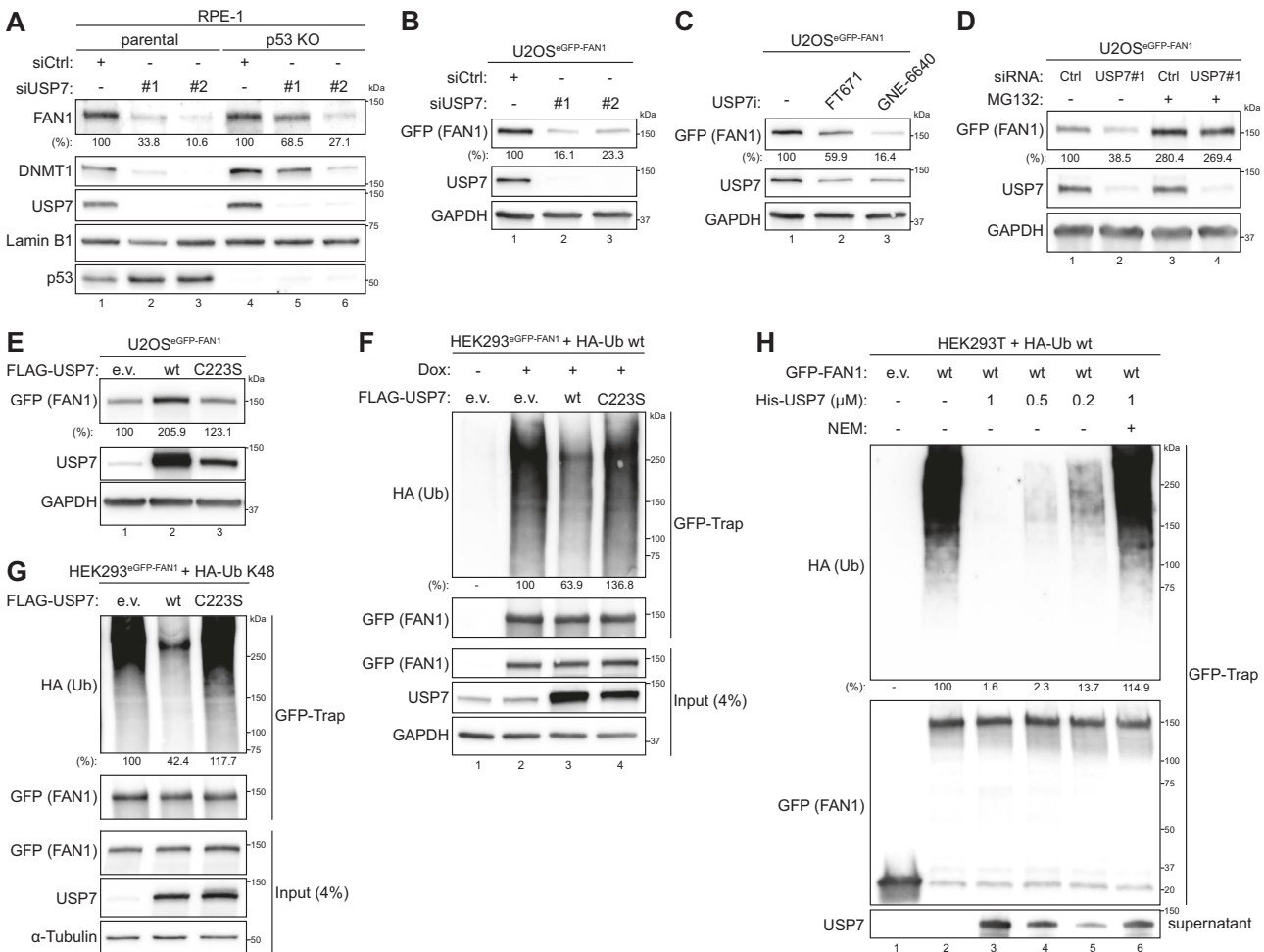

**Fig. 3 | USP7 deubiquitinates FAN1 controlling its protein levels. A** RPE-1 parental and p53 KO cell lines were transfected with either non-targeting (Ctrl) or USP7 siRNA oligos. 48 h later, whole-cell lysates were analysed by immunoblotting. **B** Doxycycline-inducible U2OS^eGFP-FAN1 cells were transfected with either non-targeting (Ctrl) or USP7 siRNA oligos. 24 h later, GFP-FAN1 expression was induced with Dox (100 ng/ml) for 24 h and whole-cell lysates were analysed by immunoblotting. **C** Same cells as in **B** were induced with Dox (100 ng/ml). 24 h later, cells were either mock-treated with DMSO (−) or treated (+) with either FT671 or GNE-6640 (10 μM) for 24 h and whole-cell lysates were analysed by immunoblotting. **D** Same cells as in **B** were transfected with either non-targeting (Ctrl) or USP7 siRNA oligos. 24 h later, GFP-FAN1 expression was induced with Dox (100 ng/ml). 24 h post induction, cells were either mock-treated (−) or treated (+) with MG-132 (10 μM) for 6 h and whole-cell lysates were analysed by immunoblotting. (**E**) Same cells as in **B** were transfected with either e.v. or indicated FLAG-USP7 expression constructs. 24 h later, GFP-FAN1 expression was induced with Dox (100 ng/ml). 24 h post induction, whole-cell lysates were analysed by immunoblotting. (**F**) Doxycycline-inducible HEK293^eGFP-FAN1 cells were co-transfected with either e.v. or HA-ubiquitin wt and either e.v. or indicated FLAG-USP7 expression constructs. 24 h later, GFP-

FAN1 expression was induced with Dox (100 ng/ml). 24 h post induction, cells were lysed and whole-cell extracts were subjected to GFP-Trap resin. Inputs and recovered protein complexes were analysed by immunoblotting. **G** Doxycycline-inducible HEK293^eGFP-FAN1 cells were co-transfected with either e.v. or HA-ubiquitin K48-only and either e.v. or indicated FLAG-USP7 expression constructs. 24 h later, GFP-FAN1 expression was induced with Dox (100 ng/ml). 24 h post induction, cells were lysed and whole-cell extracts were subjected to GFP-Trap resin. Inputs and recovered protein complexes were analysed by immunoblotting. **H** HEK293T cells were co-transfected with HA-ubiquitin wt and either e.v. or GFP-FAN1 expression constructs. 24 h later, cells were lysed and whole-cell extracts were subjected to GFP-Trap resin. Recovered GFP-Trap resin was equilibrated in DUB buffer and then incubated with the indicated concentration of purified recombinant His-USP7 in absence (−) or presence (+) of NEM (20 mM). In (**A–E**), relative FAN1 protein levels were determined by quantification of FAN1 band intensity (normalized to GAPDH or Lamin B1 loading controls) with the ImageJ software. In (**F–H**), relative HA-ubiquitin levels in the GFP-Trap were determined by quantification of HA-ubiquitin signal intensity (normalized to GFP-FAN1) with the ImageJ software. The experiments in (**A–H**) were repeated three independent times.

response to genotoxic stress[23]. We found that both siRNA-mediated depletion and catalytic inhibition of USP7 reduced MMC-induced GFP-FAN1 subnuclear foci (Figures S5A and S5B), indicating that USP7 is required for efficient retention of FAN1 at ICLs. FAN1 accumulation at sites of ICLs is dependent on FANCD2[46]. Consistent with this, depletion of either USP7 or FANCD2 markedly reduced MMC-induced FAN1 foci, with FANCD2 knockdown producing a slightly stronger effect. Notably, simultaneous depletion of both factors nearly abolished FAN1 foci formation in response to MMC, indicating that USP7 and FANCD2 regulate FAN1 accumulation at sites of DNA damage through distinct mechanisms (Fig. 4A). However, MMC-induced FANCD2 foci formation

remained intact upon USP7 depletion, suggesting that USP7 acts downstream of FANCD2 in modulating FAN1 localization (Fig. 4B).

USP7 has been shown to modulate cell survival and apoptosis in response to genotoxic stress, including treatment with DNA cross-linking agents such as cisplatin and MMC[47,48]. We observed that both depletion of USP7 as well as loss of FAN1 in HeLa cells led to increased apoptosis two days after treatment with a high dose of MMC (1 μg/ml), as indicated by elevated proteolytic cleavage of PARP-1 and caspase-3 (Fig. 4C). Apoptosis induction was further amplified in HeLa FAN1 KO cells depleted of USP7, suggesting that under these conditions of severe genotoxic stress, USP7 likely regulates multiple factors involved

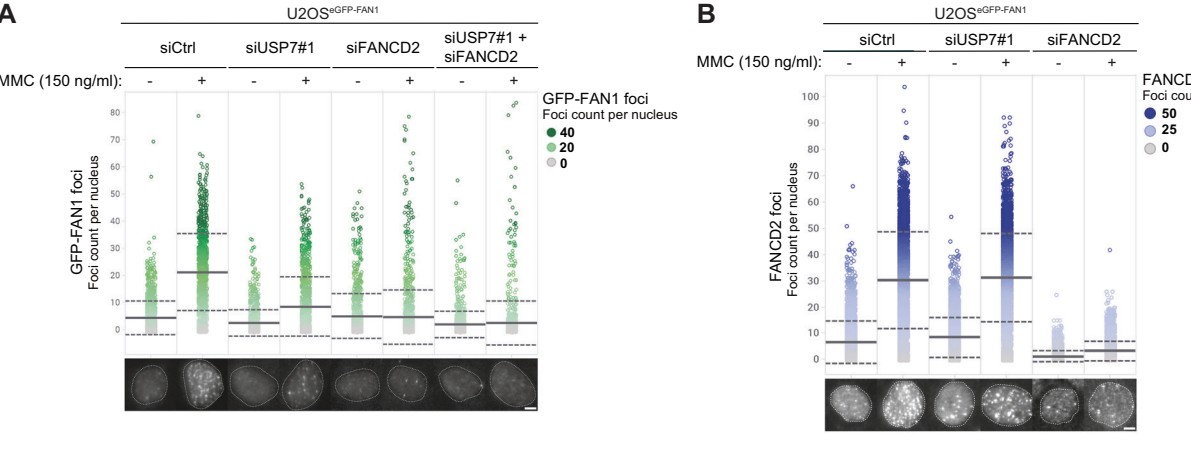

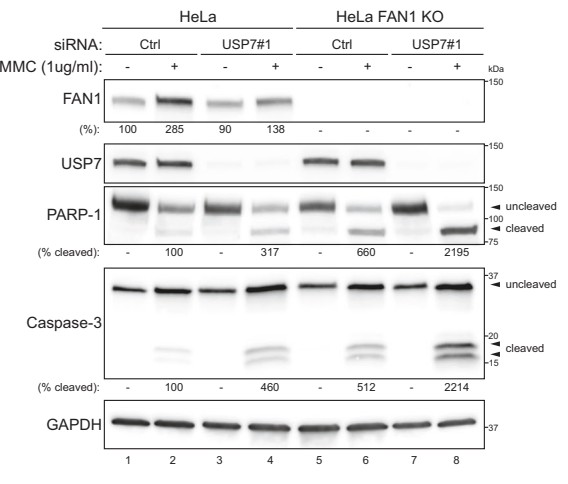

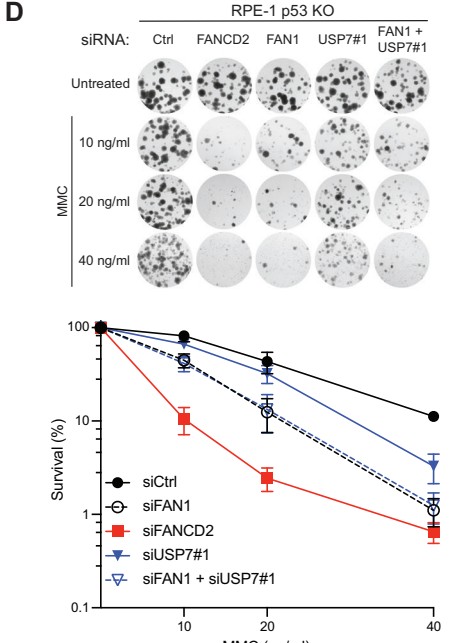

**Fig. 4 | USP7 facilitates FAN1 localization and function during ICL repair. A** QIBC analysis of chromatin-bound GFP-FAN1 foci in U2OS^eGFP-FAN1 cells transfected with either non-targeting (Ctrl), USP7 or FANCD2 siRNA oligos. 24 h later, cells were either mock-treated (−) or treated (+) with MMC (150 ng/ml) for 24 h. Color-coded scatterplots (from grey to green) indicate the number of GFP-FAN1 foci per nucleus from one biological replicate. Mean (solid line) and standard deviation (SD) from the mean (dashed lines) are indicated. Representative images are shown (bottom). Scale bar: 10 μm. This experiment was repeated three times. **B** QIBC analysis of endogenous FANCD2 foci in U2OS^eGFP-FAN1 cells transfected with either non-targeting (Ctrl) or the indicated siRNA oligos. 24 h later, cells were either mock-treated (−) or treated (+) with MMC (150 ng/ml) for 24 h. Color-coded scatterplots (from grey to blue) indicate the number of FANCD2 foci per nucleus from one biological replicate. Mean (solid line) and standard deviation (SD) from the mean (dashed lines) are indicated. Representative images are shown (bottom). Scale bar: 10 μm. This experiment was repeated three times. **C** HeLa parental and FAN1 KO cells were transfected either non-targeting (Ctrl) or USP7 siRNA oligos. 24 h later, cells were either mock-treated (−) or treated (+) with MMC (1 μg/ml) for 48 h and whole-cell lysates were analysed by immunoblotting. Arrows indicate uncleaved and cleaved forms of PARP-1 and Caspase-3, respectively. Relative FAN1, cleaved PARP-1 or cleaved Caspase-3 protein levels were determined by quantification of the respective band intensity (normalized to GAPDH) with the ImageJ software. The experiment was repeated three independent times. **D** Clonogenic survival assays of RPE-1 p53 KO cells depleted of the indicated factors (Ctrl black circle, FAN1 empty black circle, FANCD2 red square, USP7#1 inverted blue triangle, FAN1 + USP7#1, inverted empty blue triangle) and exposed to increasing doses of MMC for 24 h. Viability of untreated cells was defined as 100%. Data are presented as the means ± SD of three biological replicates (n = 3). Representative images are shown (top).

in the repair or signalling of ICL damage[49] (Fig. 4C). These findings were largely recapitulated in MMC-treated RPE-1 p53 KO cells following depletion of FAN1, USP7, or both factors simultaneously, supporting their cooperative role in response to excessive ICL damage (Figure S5C). Notably, FAN1 levels were upregulated in response to MMC, and this induction was abrogated by USP7 depletion, further supporting a role for USP7 in stabilizing FAN1 at ICLs (Fig. 4C).

To investigate the functional consequences of USP7 and FAN1 deficiency under sustained genotoxic stress, we conducted clonogenic survival assays following chronic treatment with low doses MMC at 10,

20, and 40 ng/ml. In line with previous reports, depletion of USP7 in HeLa parental cells using two independent siRNA oligos resulted in only a modest increase MMC sensitivity compared to the more pronounced effects observed with RAD18 or FANCD2 depletion[47,48,50] (Figures S5D and S5E). Notably, USP7 depletion did not exacerbate MMC hypersensitivity in HeLa FAN1 KO cells, in contrast to RAD18 or FANCD2 depletion, which led to an additive increase in sensitivity (Figures S5F and S5G). A similar epistatic relationship was observed in RPE-1 cells: the strongest MMC sensitivity was seen in FANCD2-depleted cells, while FAN1/USP7 co-depletion did not enhance

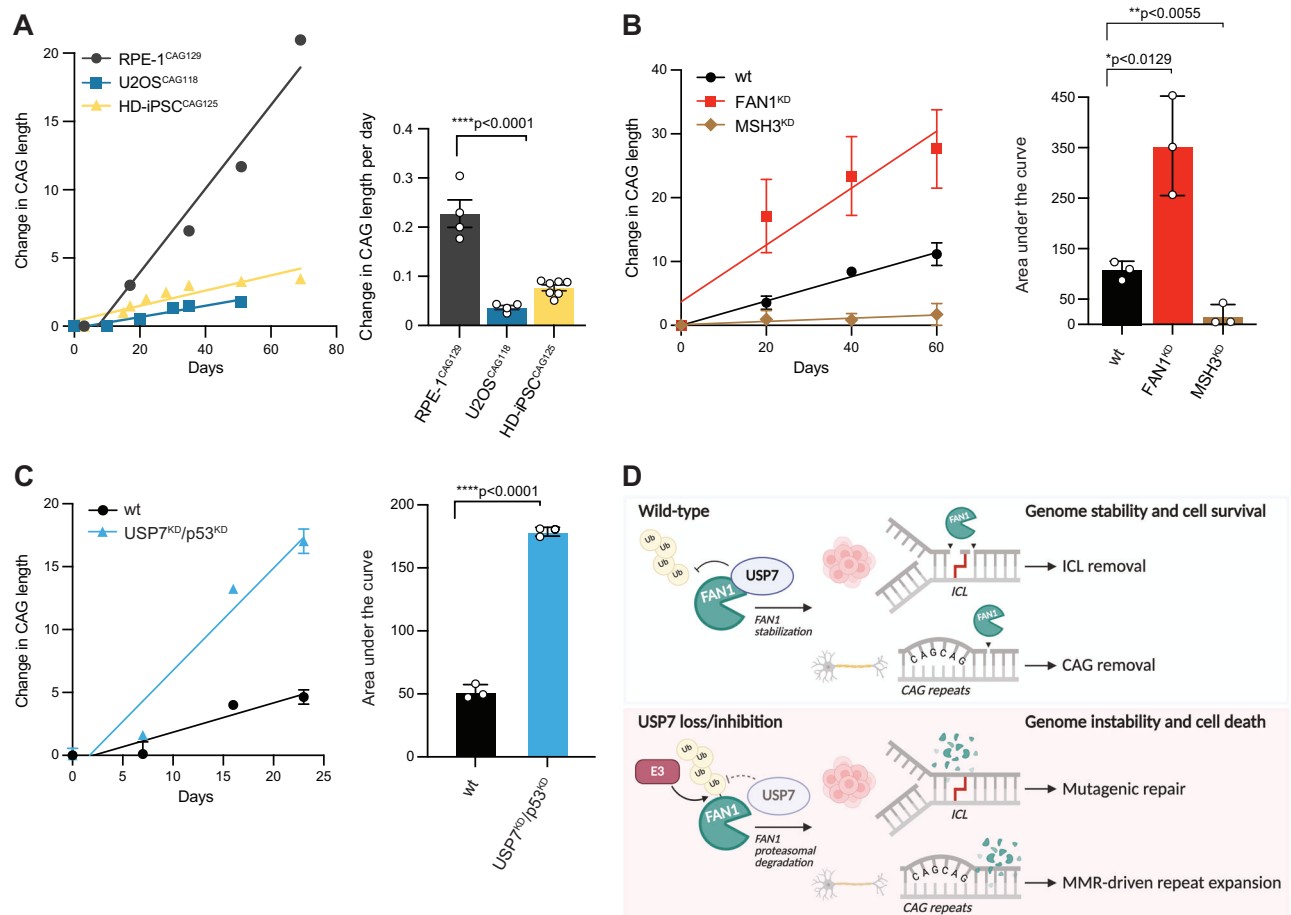

**Fig. 5 | USP7 knock-down accelerates CAG repeat expansion in RPE-1[CAG129] cells.** **A** Quantification of CAG repeat expansion across cell lines (RPE-1[CAG129], black circles; U2OS[CAG118], blue squares; HD-iPSC[CAG125], yellow triangles). Left: CAG repeat lengths were monitored over 69 days in culture ($n = 1$). Right: The RPE-1[CAG129] clone C2 exhibited the highest expansion rate compared with HD-iPSC[CAG125] and U2OS[CAG118] lines. Bar graph shows daily expansion rates. Statistical analysis was performed using one-way ANOVA followed by Tukey's multiple comparison test (****$p < 0.0001$ for RPE-1 vs. U2OS and for RPE-1 vs. HD-iPSC). Data are shown as mean ± SD; $n = 4$ for RPE1 and U2OS; $n = 7$ for HD-iPSC. **B** CAG repeat expansion over time in RPE-1[CAG129] cells following knock-down of FAN1 (red) or MSH3 (brown) compared with wild-type (wt; black) controls. Left: Line graph showing the mean change in CAG repeat length over 60 days (mean ± SD; $n = 3$ biological replicates). Right: Quantification of expansion as area under the curve (AUC). Statistical significance was determined using an unpaired two-sided $t$-test ($p = 0.0129$ for wt vs. FAN1[KD]; $p = 0.0055$ for wt vs. MSH3[KD]). Data are shown as mean ± SD; $n = 3$ biological replicates. This experiment was repeated independently three times. **C** Dual knock-down of USP7 and p53 (cyan) markedly increases CAG repeat expansion compared to wt cells (black). Left: Time-course analysis reveals a significantly higher expansion rate in double KD cells. Averages shown; error bars, mean ± SD; $n = 3$ technical replicates. Right: AUC quantification confirms the increase. Statistical significance was determined using an unpaired two-sided $t$-test (****$p < 0.0001$). Data are shown as mean ± SD, $n = 3$ technical replicates. The experiment was repeated independently three times. **D** Graphical summary depicting the consequences of USP7 loss, and subsequent FAN1 degradation, for DNA repair. Top: Under physiological conditions FAN1 is subjected to USP7-mediated deubiquitination, preventing FAN1 from being degraded by the proteasome. FAN1 promotes accurate repair of ICLs by localizing to the lesion and unhooking it, thereby conferring cellular resistance to ICL-inducing agents. Moreover, in neuronal cells, FAN1 is the main nuclease capable of nicking downstream of the CAG repeat, allowing the correct repair of slipped DNA, therefore delaying HD onset. Bottom: Loss or inhibition of USP7 results in elevated FAN1 polyubiquitination, by a yet unknown E3 ubiquitin ligase, and subsequent proteasomal degradation. Reduction of FAN1 protein levels triggers mutagenic repair of ICLs, which ultimately leads to chromosomal aberrations and cell death. In post-mitotic neurons, lack of FAN1 results in accelerated mismatch repair-dependent CAG repeat expansion, resulting in the accumulation of toxic Huntingtin, driving neuronal loss and disease progression in HD patients. Created in BioRender (https://BioRender.com/z4tedzr).

sensitivity beyond individual knock-down of FAN1 (Fig. 4D). Collectively, these findings indicate that FAN1 is the primary USP7 substrate involved in the repair of ICLs under moderate DNA damage conditions. However, under high levels of DNA crosslinking stress, the exacerbated apoptotic response in FAN1/USP7 double-depleted cells likely reflects the loss of additional USP7-regulated genome maintenance factors, such as RAD18, RNF168, or SAMHD1[47,48,51].

## USP7 depletion accelerates CAG repeat expansion

FAN1 is a key modulator of somatic CAG repeat expansion in Huntington's disease (HD), where its loss leads to increased repeat lengthening, an effect largely driven by the mismatch repair (MMR) pathway, particularly the MutSβ (MSH2–MSH3) complex[52,53]. To establish a robust system to study the somatic expansion of CAG repeats, we generated a human RPE-1 clonal cell model stably expressing *HTT*-exon1 containing 129 CAG repeats (RPE-1[HTT-CAG129]) (Figures S6A–C). In agreement with previous observations in this cellular background[54], we found that CAG repeat expansion occurred more rapidly in RPE-1[HTT-CAG129] cells than in U2OS or HD patient-derived induced pluripotent stem cell (iPSC) models (Fig. 5A). To validate this model, we performed dual-gRNA CRISPR knock-downs of either FAN1 or MSH3. As expected, FAN1 depletion increased CAG repeat expansion[10,55], while MSH3 knock-down effectively suppressed repeat instability, consistent with its essential role in driving MMR-dependent expansion[53,56] (Fig. 5B).

To determine whether USP7 modulates CAG expansion similarly to FAN1, we used pooled dual-gRNA CRISPR knock-down of USP7 in the RPE-1[HTT-CAG129] background. To bypass p53-mediated growth suppression following loss of USP7, double knock-down was performed using guides targeting both USP7 as well as p53 (USP7[KD]/p53[KD]). Effective USP7 depletion was confirmed by western blot analysis (Figure S6D) and by quantifying sgRNA cutting efficiency in pooled libraries using Sanger sequencing followed by Inference of CRISPR Edits (ICE) analysis (Figure S6E). Moreover, we found that loss of p53 alone (p53[KD]) did not affect CAG expansion relative to the wild-type cells (wt) throughout the 35-day analysis (Figure S6F). Strikingly, over a 25-day time course, USP7 knock-down led to accelerated CAG repeat expansion, mirroring the phenotype observed with FAN1 knock-down (Fig. 5B, C), suggesting that USP7 limits CAG expansion and functions in a pathway akin to FAN1. Together, these data identify USP7 as a regulator of CAG repeat stability and demonstrate the strength of the RPE-1[HTT-CAG129] model for dissecting genetic modifiers of somatic expansion in HD.

## Discussion

FAN1 is a versatile structure-specific DNA endo- and exonuclease involved in several genome stability maintenance pathways, including ICL repair[1–4], homologous recombination[1,2], replication fork integrity[23,57,58], and, most recently, in the stabilization of expanded trinucleotide repeats[12,13,59]. Despite these diverse roles, very little is known about how the activity of FAN1 is controlled in cells to safeguard genomic integrity. We previously reported that the FAN1-MLH1 interaction is cell-cycle regulated, possibly directing FAN1 nucleolytic activity to specific DNA substrates and preventing otherwise aberrant DNA cleavage[9]. Typical for a DNA repair nuclease, FAN1 expression levels are low in both cancer and non-cancerous cell lines (< 10 RNA transcripts per million, according to v.24 proteinatlas.org)[60]. Therefore, we anticipated that FAN1 protein levels are tightly controlled by the balance of E3 ubiquitin ligases and deubiquitinases. Here, we identify USP7 as a regulator of FAN1 protein stability. USP7 binds FAN1 via its N-terminal unstructured domain and prevents its proteasomal degradation through deubiquitination. Depletion of USP7 results in MMC hypersensitivity and accelerated rates of CAG repeat expansion, two phenotypes that are associated with FAN1 deficiency.

Mechanistically, we show that USP7 engages FAN1 through a bipartite interaction involving both TRAF-like and UBL domains. This dual binding mode mirrors previously reported USP7 interactions with substrates such as DNA Polymerase ι, PAF15 and ZMYND8[35,36,61]. Within the FAN1-NTD, we identified a canonical TRAF consensus-binding motif, 181-PQSS-184, as well as an upstream region (aa 154-166) that contribute to the FAN1-USP7 interaction. However, mutational analyses indicate the presence of redundant, compensatory or alternative USP7-binding sites within FAN1, consistent with a recent report describing USP7 recognition of a non-canonical RxxD motif in KRAS[62]. Despite the complexity of this interaction, we found that USP7 robustly removes K48-linked polyubiquitin chains from ubiquitinated FAN1, counteracting FAN1 proteasomal turnover.

Importantly, USP7 was shown to regulate a multitude of cellular processes, including several DNA damage response pathways[49]. Here we could show that USP7 is implicated in ICL repair, at least in part by positively regulating FAN1 activity. Indeed, depletion of USP7 resulted in a reduction in the assembly of FAN1 into subnuclear foci in response to MMC treatment. Moreover, similar to a previous study[48], we observed that USP7-depleted cells exhibit modest sensitivity to MMC. Importantly, in this context, we found that USP7 and FAN1 function in an epistatic manner, suggesting that FAN1 is a major effector of USP7 in the context of ICL repair.

Genome-wide association studies identified FAN1 as a key genetic modifier of HD onset[12,13]. Subsequently, genetic studies showed that FAN1 deficiency leads to enhanced somatic CAG expansion rates in different cellular and mouse models of HD[17,63,64]. Currently, there are different models explaining how FAN1 suppresses MMR-driven repeat expansion by competing with the MutSβ complex for the binding and subsequent cleavage of CAG extrusions and/or via its interaction with MLH1[52]. Indeed, using an RPE-1[HTT-CAG129]-based experimental system, we observed increased CAG repeat instability upon FAN1 inactivation, while repeats are stable over time in MSH3-depleted cells. We found that loss of USP7 phenocopied FAN1 loss, driving CAG expansion at comparable rates. This suggests that USP7 acts as an upstream modulator of triplet repeat stability by stabilizing FAN1. USP7 has previously been linked to polyglutamine (polyQ) neurodegenerative diseases, including HD, as it preferentially binds and potentially stabilizes polyQ-expanded mHTT and androgen receptor[65]. Moreover, individuals carrying pathogenic variants in USP7 are affected by the Hao-Fountain Syndrome (HAFOUS), a neurodevelopmental disorder characterized by global developmental delay, intellectual disability, severe speech delay and behavioural abnormalities[66]. Thus, while our data suggests a potential role of USP7 in preventing somatic repeat expansions in HD brains by stabilizing FAN1 protein levels, we cannot exclude additional functions of USP7 in regulating trinucleotide repeat (TNR) metabolism through additional substrates.

Emerging evidence suggests that trinucleotide repeat length may influence cancer susceptibility, with longer repeats, such as the expanded CAG tract in exon 1 of mutant huntingtin (mHTT), associated with reduced cancer incidence in HD cohorts[67,68]. Our findings place FAN1 at a mechanistic crossroads: USP7-dependent FAN1 stabilisation maintains genome integrity in proliferating cells while limiting repeat expansion in post-mitotic neurons. Consequently, USP7 inhibition could have context-dependent effects, enhancing chemosensitivity in p53-deficient tumours, yet potentially accelerating TNR instability in the nervous system, highlighting the USP7-FAN1 axis as a molecular link between cancer biology and repeat expansion disorders. These insights also carry therapeutic implications. FAN1 protects cells from ICL-induced cell death, suggesting that USP7 inhibition could sensitize cancers to DNA cross-linking agents[69]. Conversely, enhancing USP7 activity to stabilize FAN1 may benefit carriers of TNR-associated diseases. Supporting this concept, Cheng and colleagues showed that transfection of codon-optimized FAN1 mRNA via lipid nanoparticles (LNPs) in HD iPSC-derived astrocytes blocked CAG repeat expansion; however, delivery in HD mice yielded only transient FAN1 upregulation[70]. These findings underscore the need for alternative strategies, including the development of USP7-based deubiquitinase-targeting chimeras (DUBTACs) specific for FAN1[71].

## Methods
### Cell culture
U2OS, HEK293, HEK293T, HeLa and RPE-1 cells [American Type Culture Collection (ATCC)] were grown in Dulbecco's modified Eagle's medium (DMEM) supplemented with 10% fetal calf serum (FCS), penicillin (100 U/mL) and streptomycin (100 μg/mL). U2OS Flp-In T-REx, HEK293 Flp-In T-REx and HeLa Flp-In T-REx (Invitrogen, Life Technologies) cells were maintained in medium supplemented with blasticidin (10 μg/ml) and hygromycin B (250 μg/mL). The Flp-In T-REx system (Thermo Fisher Scientific) was used to generate cell lines stably expressing different short hairpin RNA (shRNA)-resistant forms of eGFP-FAN1 constructs under the control of a doxycycline (Dox)-inducible promoter as previously described[9]. RPE1-Cas9-CAG118 were maintained at 37 °C in 5% CO2 and 5% O2. The same cells were cultured in high-glucose DMEM supplemented with 10% fetal bovine serum (FBS), 1% glutamine, 1% non-essential amino acids, penicillin (100 U/mL) and streptomycin (100 μg/mL). HEK293T used for viral production were cultured in high-glucose DMEM, supplemented with 10% fetal bovine serum (FBS), 1% non-essential amino acids, 1% sodium pyruvate and 1% GlutaMAX.

## Virus Production

6-well plates were coated with 0.1% (w/v) gelatine in PBS for 5 minutes. HEK293T cells were seeded at approximately 80% confluency. On day 2, cells were transfected using Lipofectamine™ LTX (Invitrogen) and Opti-MEM™ (Invitrogen). The transfection mixture included the lentiviral transfer vector (targeting USP7, p53, FAN1, or MSH3), packaging plasmid psPAX2, and envelope plasmid pMD2.G, prepared according to the manufacturer's instructions and incubated for 5 minutes at room temperature after addition of the PLUS reagent. Lipofectamine LTX was then added, and the solution incubated for 30 minutes. Meanwhile, the medium was aspirated from the HEK293T cells, and 5 mL of fresh medium was added. 3 mL of the transfection mixture was added dropwise to each dish. On day 3, the medium was replaced with 8 mL of fresh medium and cells were incubated for an additional 48 hours. On day 6, the virus-containing supernatant was harvested, centrifuged at 1200 g for 3 minutes to remove cell debris, and filtered through a 0.45 μm low protein-binding SFCA syringe filter (Nalgene). The virus was aliquoted into 1 mL tubes and stored at −80 °C for up to four months.

## CRISPR-Cas9 gene editing of RPE-1 cells

To generate knock-out cells, RPE1-Cas9-CAG118 cells were seeded at a density of 100,000 cells/mL, and 1:1000 polybrene was added to the mixture. The prepared virus was introduced into the wells at a multiplicity of infection (MOI) of 80% for the following targets: USP7, p53, FAN1, and MSH3. After 3 days, the MOI was assessed using FACS. Cells were selected by treatment with 5 μg/mL puromycin for 3 days. The cells recovered over the course of approximately 7 days. Once the cells reached confluence, samples were collected every 7 days, for a total of 6 time points, with the media being replaced every 4 days. CRISPR cutting efficiency was assessed using the Synthego ICE (Inference of CRISPR Edits) analysis tool[72]. Sanger sequencing chromatograms of the target loci were uploaded to the online platform, which deconvolves mixed trace signatures to quantify indel frequencies and infer editing outcomes. The tool provided ICE scores, indel spectra, and predicted % knockout efficiency, allowing rapid and reliable evaluation of CRISPR editing across experimental conditions.

## Generation of a HeLa FAN1 knock-out cell line

HeLa FAN1 knock-out cells were generated using the CRISPR/Cas9 system. hSpCas9-2A-Puro plasmids (Addgene, nr. 62988) were constructed, each containing the Cas9 protein and a specific guide RNA (gRNA) targeting exon 1 of the FAN1 gene. The two gRNA target sequences (listed in Table S1) were 250 bp apart. Cells were transfected in tandem with both plasmids using Lipofectamine 2000 (Thermo Fisher Scientific) according to the manufacturer's protocol. Following transfection, cells were selected with puromycin (2 μg/mL) for 48 h to enrich for successfully transfected populations. Individual clones were then isolated and expanded. Clones were screened for CRISPR-mediated genome editing at the target region by sequencing of genomic DNA and knock-out was confirmed by western blot analysis.

## Bacterial strains

Chemo-competent XL1-Blue E. coli were used for site-directed mutagenesis and chemo-competent Stellar™ cells (TakaraBio) for plasmid cloning using the In-Fusion® technology (TakaraBio). Recombinant GST-FAN1 was expressed in electro-competent BL21-CodonPlus_RIL E. coli. Human recombinant full-length USP7 was purchased from R&D Systems (# E-519).

## Bacterial and mammalian expression vectors

All primers used for cloning and site-directed mutagenesis are listed in Table S2. Site-directed mutagenesis was carried out by PCR using CloneAmp™ HiFi PCR Premix (TakaraBio) according to the manufacturer's instructions and subsequent digestion of template DNA with DpnI (NEB). pQFlag-USP7 was a gift from Goedele Maertens & Gordon Peters (Addgene # 46751). pcDNA5.1 FRT/TO GFP-FAN1 deletion constructs were kindly shared by Sarah J. Tabrizi (UCL, UK). The PCR product of the USP7 TRAF (residues 2-207) domain was cloned into a pNICBio2 vector containing N-terminal His-tag and C-terminal AviTag through ligation-independent cloning (LIC) using In-Fusion seamless cloning (Takara). The PCR product of the UBL domain of USP7 (residues 520-1067) was cloned into a p28BIOH-LIC vector containing N-terminal AviTag and C-terminal His-tag through LIC using In-Fusion seamless cloning (Takara).

## Purification of recombinant human USP7 fragments

All plasmids were transformed into BL21(DE3)-pBirAcm cells. The cells were grown using a Large-scale EXpression (LEX) bioreactor at 37 °C. Once OD600 reached 0.7, the temperature was reduced to 15 °C. At OD600 0.8-1, expression of biotinylated proteins was induced by adding 25 mg/L biotin and 0.5 mM isopropyl β-d-1-thiogalactopyranoside (IPTG), and growth continued overnight. Cell cultures were harvested by centrifugation at 6500 g, 10 min, 10 °C (JLA8.1000). Pellets were flash-frozen in liquid nitrogen and stored at −80 °C prior to purification. Cell pellets were thawed, diluted with pre-chilled resuspension buffer (20 mM HEPES pH7.4, 500 mM NaCl, 5% (v/v) glycerol, 1 mM TCEP) and supplemented with 3 μg/ml benzonase, 2 mM MgSO4 and 0.8 mM PMSF/benzamidine-HCl. The mixture was homogenized and lysed by sonication, 60 cycles of 5 s pulse followed by 7.5 s rest. Cell lysates were clarified by centrifugation at 40000 g, 1 hr, 10 °C (JLA16.250). The supernatant was incubated with TALON resin slurry (2 mL/L pellet) (Takara, #635504) for 30 minutes on an end-to-end shaker at 4 °C. Affinity purification was performed using a gravity column. Two wash steps were performed using resuspension buffer, first without and next with 5 mM imidazole. The protein was eluted using 300 mM imidazole. The protein was concentrated using Amicon centrifugal filter units (10 kDa and 50 kDa for TRAF and UBL domains respectively) then loaded onto a HiLoad 16/60 Superdex200 prep grade column (Cytiva) for size-exclusion chromatography on an AKTA pure purification system (Cytiva) using gel filtration buffer (GFB) (20 mM HEPES pH7.4, 300 mM NaCl, 2.5% (v/v) Glycerol, 1 mM TCEP). Peak fractions were concentrated. The protein was flash frozen in liquid nitrogen and stored in −80 °C in GFB. Protein purity was analysed by SDS-PAGE and identity was confirmed by mass spectrometry. Biotinylation of proteins were confirmed by streptavidin (Invitrogen, #434301) gel shift assay.

## Purification of recombinant human FAN1

Transformed clones of chemocompetent BL21 bacteria were picked and inoculated in LB medium supplemented with 50 μg/ml kanamycin, subsequently incubated at 37 °C overnight under agitation at 250 rpm. The following day, the 5 ml cultures were transferred into a larger Erlenmeyer flask, inoculated again at a 1:50 ratio with LB-medium and shaken at 30 °C until the optical density at wavelength of 600 nM (OD600) reached a value between 0.7 and 0.8. Bacteria were allowed to reach a temperature of 18 °C before inducing protein expression by addition of 0.5 mM isopropyl β-D-thiogalactoside (IPTG) (Sigma-Aldrich), with shaking at 18 °C and 250 rpm overnight. Bacteria were collected by centrifugation at 4 °C and 4500 g for 30 min, and the pellet was weighted and resuspended in ice cold lysis buffer (25 mM Tris-HCl pH 8.0, 300 mM NaCl; 1 ml per 4 g of pellet). 1 mM PMSF, protease inhibitor cocktail (cOmplete, EDTA- free, Sigma), and 0.1 mg/ml lysozyme were added, and the mixture stirred at 4 °C for 30 min. Samples were then sonicated on ice (70% amplitude, 10 s on 50 s off, 10-minute cycles). 2 μg/ml of Benzonase was added and the samples were incubated rotating at 4 °C for 30 min. After a spin-down at 40000 g for 1 hour at 4 °C, the insoluble material was removed. Induced proteins were purified from soluble extracts by affinity chromatography using 5 ml HisTrap™ column and proteins eluted

using a 20 to 200 mM Imidazole gradient. Elution fractions containing the protein of interest were pooled and purified further by a 5 ml MBPTrap HP column. Elution was performed using a 20 mM maltose elution buffer. Fractions containing the purified protein were additionally run on a HiPrep™ 26/10 Desalting column and purified recombinant full length His6-MBP-FAN1-His6 was stored in 20 mM Tris-HCl pH 8, 150 mM NaCl and 2 mM β-Mercaptoethanol. Purity of the protein was assessed by boiling in SDS sample buffer and separating by SDS-PAGE followed by staining using Coomassie Blue.

### siRNA transfections

Small interfering RNA (siRNA) oligos were purchased from Microsynth and transfected at a final concentration of 40 nM using Lipofectamine RNAiMAX (Thermo Fisher Scientific). A list of all siRNA oligos used throughout this study can be found in Table S3.

### Antibodies

A complete list of all primary antibodies used throughout this study can be found in Table S4. Secondary horseradish peroxidase (HRP)-conjugated anti-mouse and anti-rabbit antibodies were from GE Healthcare, and the HRP-conjugated anti-goat antibody was purchased from Santa Cruz Biotechnology. Alexa Fluor 488-conjugated secondary antibody was purchased from Thermo Fisher Scientific. The monoclonal antibody against FAN1 was produced by GenScript (NJ, USA) by immunizing mice with recombinant His-tagged FAN1 (aa 871 to 1017) purified from *E. coli*.

### Chemicals and peptides

4′,6-diamidino-2-phenylindole (DAPI), Mitomycin C (MMC), cycloheximide (CHX) and MG132 were purchased from Sigma-Aldrich. FT671 and GNE-6640 were purchased from Aobius. TAK-243 was purchased from MedChemExpress. Doxycycline was purchased from TaKaRa Clontech. A complete list of all peptides used throughout this study can be found in Table S5. The synthetic biotinylated FAN1 peptides used in this study were purchased from SynPeptide (Shanghai, China) at 95% purity. The synthetic FITC-labelled FAN1 peptides used in this study were purchased from GenScript (NJ, USA) with a > 95% purity. Custom-designed FAN1 60-mer peptide (aa 118–177) was purchased from GenScript (NJ, USA) and synthesized with a purity >85%.

### Affinity purification coupled to mass spectrometry

HEK293 Flp-In T-REx$^{eGFP-FAN1}$ cells were used to prepare nuclear protein-enriched extracts. Parental HEK293 were used as negative control. After incubation in Swelling Buffer (10 mM Tris-HCl pH 8, 10 mM NaCl) for 10 min at 4 °C, cells were collected and centrifuged at 1000 g for 10 min at 4 °C. Nuclei were resuspended in IP Buffer (50 mM Tris-HCl pH 7.5, 100 mM NaCl, 1 mM MgCl$_2$, 10% Glycerol, 0.1% NP40), supplemented with protease inhibitor cocktail (cOmplete Ultra, EDTA-free, Sigma), treated with Benzonase (100 U) for 15 min at 4 °C and disrupted by syringing using a 27 G needle. After 21000 g centrifugation for 5 min at 4 °C, supernatants containing 1 mg of nuclear enriched proteins were incubated with 25 μL of ChromoTek GFP-Trap® Agarose resin (Proteintech) for 1 h at 4 °C. Beads were washed 4 times in IP buffer, once in IP buffer without NP40 and 4 times in 100 μL of 1X PBS before on-beads tryptic digestion. Mass spectrometry analysis (direct LC/MS-MS) was performed using a high-resolution timsTOF Pro (Bruker) coupled to an Evosep One (Evosep). Samples were separated with the extended Evosep method "15 samples/day" keeping the analytical column (PSC-15-100-3-UHPnC, ReproSil C18 3 m 120 Å 15 cm ID 100 μm, PepSep) at 50 °C. For the dual timsTOF, MS spectra were scanned from m/z 100 to m/z 1700 in ddaPASEF mode (data dependent acquisition Parallel Accumulation Serial Fragmentation). For the ion mobility settings, the inversed mobilities from 1/K0 0.60 Vs/cm2 to 1.60 Vs/cm2 were analysed with ion accumulation and ramp time of 166 ms, respectively. 1 survey TIMS-MS scan is followed by 10 PASEF ramps for

MS/MS acquisition, resulting in a 1.9 s cycle time. Singly charged ions are excluded using the polygon filter mask and isolation windows for MS/MS were set to m/z 2.0 for precursor ions below m/z 700, and m/z 3.0 for ions above. Then, the acquired MS data were processed using the Fragpipe v17 (MSFragger search engine). The spectra were searched using the Homo sapiens (UP000005640) database with the variable modifications Acetyl (Protein N-term) and Oxidation (M), and the fixed modification Carbamidomethyl (C). Proteomics differential expression analysis was performed with the R-package prolfqua using default settings[73].

### Pull-down assays

For GST pull-down assays, GST fusion plasmids were transformed in BL21 RIL (CodonPlus) *E. coli* (Stratagene) and recombinant proteins were expressed by incubating the bacteria for 24 h at 16 °C after the addition of 100 μM IPTG. After centrifugation, the bacterial pellet was resuspended in cold PBS, supplemented with 1% Triton X-100 and protease inhibitors (1 mM PMSF, 1 mM benzamidine, and cOmplete™ Protease Inhibitor Cocktail). After sonication and centrifugation, GST-tagged FAN1 was purified from soluble extracts using Glutathione Sepharose 4 Fast Flow beads (GE Healthcare). GST fusion proteins bound to glutathione beads were mixed with 0.5 μg of human recombinant USP7 (R&D Systems) and incubated for 1 h at 4 °C in 1 ml of PBS-1% Triton. Beads were then washed three times with NTEN300 buffer (0.5% NP-40, 0.1 mM EDTA, 20 mM Tris-HCl pH 7.4 and 300 mM NaCl) and once with TEN100 (20 mM Tris-HCl pH 7.4, 0.1 mM EDTA and 100 mM NaCl) buffer. Recovered complexes were boiled in SDS sample buffer and analysed by SDS–PAGE followed by immunoblotting.

For peptide pull-down assays, biotinylated FAN1 peptides were incubated with Strep-Tactin® XT 4Flow® resin (IBA LifeScience) at room temperature for 30 minutes and washed four times with 0.1% Triton-PBS and two times with HNE buffer (20 mM HEPES pH 7.5, 0.2 mM EDTA, 0.5% NP-40, 150 mM NaCl, 0.5 mM DTT, 0.1% BSA, 0.5 mM PMSF). Beads were then mixed with 0.5 μg of human recombinant USP7 (R&D Systems) and incubated rotating for 2 h at 4 °C. Beads were then washed three times with HNE buffer, once with TEN300 (20 mM Tris-HCl pH 7.4, 0.1 mM EDTA and 300 mM NaCl) and once with TEN100 buffer. Complexes were boiled in SDS sample buffer and analysed by SDS–PAGE followed by immunoblotting.

### Co-immunoprecipitation

Cells were lysed in NP-40 buffer [50 mM tris-HCl (pH 7.5), 120 mM NaCl, 1 mM EDTA, 6 mM EGTA, 15 mM sodium pyrophosphate, and 1% NP-40, supplemented with phosphatase inhibitors (20 mM NaF and 1 mM Na$_3$VO$_4$) and protease inhibitors (1 mM benzamidine, 0.1 mM PMSF and cOmplete™ Protease Inhibitor Cocktail)], incubated with Benzonase (Merck) for at least 30 min at 4 °C, and clarified by centrifugation. Between 1 and 3 mg of lysates were incubated with anti-FLAG M2 (Sigma-Aldrich) or ChromoTek GFP-Trap® Agarose resin (Proteintech) for 2 h or overnight at 4 °C. The beads were then washed three times with NP-40 buffer and once with TEN100 buffer. Retrieved protein complexes were boiled in SDS sample buffer and analysed by SDS-PAGE followed by immunoblotting.

### FAN1 immunoprecipitation

5 mg of HeLa nuclear extract were pre-cleared for 2 hours at 4 °C with rotation. Two micrograms of anti-FAN1 rabbit polyclonal antibody were added to the sample and incubated overnight at 4 °C with rotation. Protein A Sepharose beads (CL4B, Sigma-Aldrich) were equilibrated in HeLa nuclear extracts buffer [20 mM HEPES (pH 7.9), 100 mM KCl, 0.2 mM EDTA, 20% Glycerol, 0.5 mM PMSF, 0.5 mM DTT], and 30 μl of bead slurry was added to each sample and incubated for 2 hours at 4 °C with rotation. The beads were washed three times with HeLa nuclear extracts buffer and once with TEN100 buffer. Retrieved

protein complexes were boiled in SDS sample buffer and analysed by SDS-PAGE followed by immunoblotting.

## Ubiquitination assay

HEK293 Flp-In T-REx$^{eGFP-FAN1}$ cells were co-transfected with HA-ubiquitin and FLAG-USP7 expression constructs, respectively. 48 h after transfection, cells were scraped in 500 µl of RIPA buffer (50 mM Tris-HCl, pH 7.5, 150 mM NaCl, 1% IGEPAL, 0.5% sodium deoxycholate, 0.1% SDS) supplemented with phosphatase inhibitors (20 mM NaF and 1 mM Na$_3$VO$_4$), protease inhibitors (1 mM benzamidine and 0.1 mM PMSF) and the deubiquitinases inhibitor N-ethylmaleimide (NEM, 20 mM). Between 1 and 3 mg of lysates were incubated with ChromoTek GFP-Trap® Agarose resin (Proteintech) for 2 h or overnight at 4 °C. The beads were then washed once with RIPA buffer, two times with NTEN 500 (0.5% NP-40, 0.1 mM EDTA, 20 mM Tris-HCl pH 7.4 and 500 mM NaCl), once with TEN300 and once with TEN100 buffer. Retrieved protein complexes were boiled in SDS sample buffer and analysed by SDS-PAGE followed by immunoblotting.

## In vitro deubiquitination assay

HEK293T cells were co-transfected with plasmids encoding HA-ubiquitin wt and GFP-FAN1 wt, respectively. 24 h after transfection, cells were washed twice with cold PBS and directly lysed in 1 ml of RIPA buffer (50 mM Tris-HCl pH 7.5, 150 mM NaCl, 1% IGEPAL, 0.5% sodium deoxycholate, 0.1% SDS) supplemented with 2 mM MgCl$_2$, cOmplete inhibitor cocktail (Roche), phosphoSTOP (Roche), 25U/ml benzonase, 0.1 mM PMSF and 20 mM N-ethylmaleimide (NEM). Lysates were incubated for 5 min at room temperature and then centrifuged at 15000 g for 15 min. Between 1 and 3 mg of lysates were incubated with ChromoTek GFP-Trap® Agarose resin (Proteintech) for 2 h or overnight at 4 °C. Beads were then collected by centrifugation, washed once with RIPA buffer, two times with NTEN 500 (0.5% NP-40, 0.1 mM EDTA, 20 mM Tris-HCl pH 7.4 and 500 mM NaCl), once with TEN300 and four times with TEN100 buffer. Beads were then equilibrated in DUB buffer (50 mM Tris-HCl, pH 7.5, 50 mM NaCl, 1 mM MgCl$_2$, 1 mM DTT), equally distributed in reaction tubes and incubated with purified recombinant His-USP7 for 2 h at 37 °C at low rpm. The unbound fraction (supernatant) was separated from the affinity resin following centrifugation, and the reaction was blocked by the addition of 5x SDS sample buffer. Samples were boiled and analysed by SDS-PAGE followed by immunoblotting.

## Immunoblotting

If not stated otherwise, cells were lysed in RIPA buffer (50 mM Tris-HCl, pH 7.5, 150 mM NaCl, 1% IGEPAL, 0.5% sodium deoxycholate, 0.1% SDS) supplemented with phosphatase inhibitors (20 mM NaF and 1 mM Na$_3$VO$_4$) and protease inhibitors (1 mM benzamidine and 0.1 mM PMSF), incubated with Benzonase (Merck) for at least 30 min at 4 °C, and clarified by centrifugation. Proteins were resolved by SDS-PAGE and transferred to nitrocellulose membranes. Immunoblots were performed with the indicated antibodies, and proteins were visualized using the Advansta WesternBright ECL reagent and the Fusion Solo S imaging system. Relative protein levels were determined by quantification of the protein of interest band intensity normalized to the loading control with the ImageJ software.

## Fluorescence polarization assay

USP7 TRAF (aa 2-207) was serially diluted (14-point, 2-fold) in assay buffer (20 mM HEPES (pH 7.4), 50 mM KCl, 1 mM TCEP, 0.005% Tween-20 (v/v), 0.2% DMSO (v/v)) from a top concentration of 200 µM. 1 µL of 200 nM N-terminally FITC-labelled FAN1-derived peptide was added to 19 µL of serially diluted protein in triplicate in a low volume flat bottom black polystyrene 384 well plate (Corning, REF 3820). The plate was covered with an aluminium foil seal and centrifuged at room temperature for 1 minute at 100 g. Fluorescence polarization (mP, milli-polarization units) was measured using Agilent BioTek Synergy H1 with excitation 485 nm, emission 528 nm, gain 80, in a kinetic read (6 measurements in 30 min). The average of 6 measurements were transformed relative to baseline FP (0 µM USP7 TRAF) and visualized in GraphPad Prism. The FP was fit using Specific binding with Hill slope.

## Colony formation assays

HeLa and RPE-1 p53 KO cells were transfected with the indicated siRNA. The next day, the cells were seeded in six-well plates. The next day, the cells were treated with MMC for 24 h. Plates were then washed twice with 2 ml of PBS before adding fresh growth medium to the cells, that were then cultured for 13 days at 37 °C before fixation with crystal violet solution [0.5% crystal violet and 20% ethanol (w/v)]. For analysis, plates were scanned and analysed with the ImageJ Plugin ColonyArea using the parameter "intensity percent".

## Flow cytometry analysis

RPE-1 parental and p53 KO cells were transfected with the indicated siRNA. 48 h later, cells were incubated with 10 µM 5-Ethynyl-2′-deoxyuridine (EdU) for 30 min at 37 °C. Cells were then harvested by trypsinization, washed, and fixed in 4% Formaldehyde in PBS. EdU labelling was carried out using the Click-iT EdU technology (Thermo Fisher Scientific) as described in the manufacturer's instructions. DNA was stained by incubating the cells in 1 % BSA/PBS containing 0.1 mg/ml RNase and 1 µg/ml DAPI. A minimum of 20,000 events were recorded with an Attune NxT flow cytometer.

## Immunofluorescence microscopy

For high-content microscopy and QIBC analyses, cells were seeded on sterile 12 mm glass coverslips and allowed to proliferate until they reached a cell density of 70 – 90%. Cells were then washed once with PBS before fixation in 3% formaldehyde in PBS for 15 min at room temperature, washed once in PBS, permeabilized for 5 min at room temperature in 0.2% Triton X-100 in PBS, washed twice in PBS, and incubated in blocking solution (filtered DMEM containing 10% FBS and 0.02% Sodium Azide) for 15 min at room temperature. When the staining was combined with an EdU Click-iT reaction, this reaction was performed prior the incubation with primary antibody according to the manufacturer's recommendations (Thermo Fisher Scientific). Where indicated, cells were pre-extracted in 0.2% Triton X-100 in PBS for two minutes on ice prior fixation. All primary antibodies were diluted in blocking solution and incubated for 2 h at room temperature. Secondary antibodies (Alexa Fluor 488, 568, 647 anti-mouse and anti-rabbit IgG from Thermo Fisher Scientific) were diluted 1:500 in blocking solution and incubated at room temperature for 1 h. Cells were washed once with PBS and incubated for 10 min with 4′,6-Diamidino-2-Phenylindole Dihydrochloride (DAPI, 0.5 mg/ml) in PBS at room temperature. Following three washing steps in PBS, coverslips were briefly washed with distilled water and mounted on 5 ml Mowiol-based mounting media (Mowiol 4.88 (Calbiochem) in Glycerol/TRIS).

## Quantitative image-based cytometry (QIBC)

Automated multichannel wide-field microscopy for QIBC was performed using the Olympus ScanR System as described previously[74]. Images of cell populations were acquired under non-saturating conditions with Olympus ScanR Image Acquisition software 3.2 and 3.3.0, typically 16 (4×4) images per coverslip, and identical settings were applied to all samples of the same experiment. Images were analysed with the inbuilt Olympus ScanR Image Analysis Software Version 3.2 and 3.3.0, a dynamic background correction was applied, and nuclei segmentation was performed using an integrated intensity-based object detection module based on the DAPI signal. All downstream analyses were focused on properly detected nuclei containing a 2C-4C DNA content as measured by total and mean DAPI intensities. Fluorescence intensities were quantified and are depicted as arbitrary units. Color-coded scatterplots of asynchronous cell populations were

generated with Spotfire data visualization software (TIBCO Spotfire 10.10.1.7). Within one experiment, similar cell numbers were compared for the different conditions. For visualizing discrete data in scatterplots, mild jittering (random displacement of data points along discrete data axes) was applied to demerge overlapping data points. Representative scatterplots and quantifications of independent experiments, typically containing several thousand cells each, are shown.

### In situ proximity ligation assay

HeLa parental and FAN1 KO cells were grown in the absence or presence of MMC (150 ng/ml) for 24 hours. Cells were pre-extracted with CSK buffer containing 0.5% of Triton™ X-100 (Sigma–Aldrich) for 5 min on ice and fixed in 4% formaldehyde in PBS (w/v) for 20 min at room temperature (RT). Coverslips were then washed with PBS and stored overnight at 4 °C. In situ PLA was performed using Duolink PLA technology (Sigma-Aldrich) according to the manufacturer's instructions. In brief, coverslips were blocked for 30 min at 37 °C with blocking solution and then incubated with the respective primary antibodies for 2 h at 37 °C. Coverslips were washed three times for 5 min in Wash Buffer A (0.01 M Tris, 0.15 M NaCl, and 0.05% Tween 20). Then, Duolink anti-Mouse PLUS and anti-Rabbit MINUS PLA probes were coupled to the primary antibodies for 1 h at 37 °C. After three wash steps in Wash buffer A for 5 min, the PLA probes were ligated for 30 min at 37 °C. Coverslips were then washed three times for 5 min in Wash buffer A. Amplification using the "Duolink In Situ Detection Reagents FarRed" (Sigma-Aldrich) was performed at 37 °C for 100 min. After amplification, coverslips were washed twice in Wash Buffer B (0.2 M Tris and 0.1 M NaCl) for 10 min and incubated for 30 min at 37 °C with the appropriate secondary antibodies. Coverslips were then washed twice with Wash Buffer B and once in 0.01× Wash Buffer B for 1 min. Lastly, the coverslips were mounted using Vectashield Mounting Media (Vector Laboratories) containing DAPI, sealed and imaged on a Leica DMI 6000 fluorescence microscope at x63 magnification. Analysis of PLA signals was performed using CellProfiler™.

### CAG-length measurement

Genomic DNA was isolated from each sample using the DNeasy Blood & Tissue Kit (QIAGEN). The expanded CAG tract was then amplified by PCR, using 150 ng of template DNA and the following primers: 6-FAM–labelled forward primer HD3F and reverse primer HD5. Thermocycling conditions were 95 °C for 10 min; 30 cycles of 95 °C for 30 s, 58 °C for 30 s and 72 °C for 90 s; followed by a final extension at 72 °C for 7 min. PCR products were then run on an ABI 3730xl Genetic Analyzer (Thermo Fisher Scientific) with the MapMarker 1000 ROX size standard (Tebubio). GeneMapper (Thermo Fisher Scientific) was then used for the alignment of the size standard for all samples. The median, standard deviation, and the instability index were calculated using the, a custom program (Romeo package) available at https://michaelflower.org.

### Reporting summary

Further information on research design is available in the Nature Portfolio Reporting Summary linked to this article.

## Data availability

All data are available within the Article, Supplementary Information, or Source data file. Raw data used to generate all graphs and derived statistics are provided in the Source data file. Original, uncropped blots can also be found in the Source data file. All original microscopy images will be made available upon request. The mass spectrometry data generated for this study have been deposited in the PRIDE database via PXD identifier PXD065101[75]. Source data are provided with this paper.

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

## Acknowledgements

We thank Lorenza Penengo for helpful discussions and providing plasmids for tagged-ubiquitin expression. We thank Sarah Tabrizi and Rob Goold for providing GFP-FAN1 expression constructs. We gratefully acknowledge Mike Flower, Lucy Coupland, and the Tabrizi Lab for generously providing the CAG expansion tools and their invaluable expertise in somatic instability analyses. We additionally thank Denny Yang Tze Te and Kangning He for their assistance with the ICE analysis used to verify the editing outcome in the MSH3 knockdown line. We gratefully acknowledge the Functional Genomics Center Zurich (FGCZ) of University of Zurich and ETH Zurich, for the support on proteomics analyses. This work was supported by research grants from the Swiss National Science Foundation (31003A_176161 and 310030_208143 to A.A.S.) and the Worldwide Cancer Research (grant reference number: 23-0355 to A.P. and A.A.S.). Work in the Balmus laboratory is supported by the UK Dementia Research Institute through UK DRI Ltd, principally funded by the UK Medical Research Council as well as CHDI Foundation, the Romanian Ministry of Research, Innovation, and Digitization (grant #PNRR-III-C9-2022-I8-66, contract 760114) and the Hereditary Disease Foundation.

## Author contributions

G.C. performed most cloning, interaction studies, western blot experiments, cell cycle analysis, and colony survival assays under the supervision of A.P. and A.A.S. E.R. and S.B. prepared mass spectrometry samples and analysed the proteomics data. F.V. performed in situ-PLA experiments. D.G.V generated HeLa FAN1 KO cells and prepared recombinant human FAN1. K.M.F. prepared recombinant GST-FAN1 fragments. M.G. performed QIBC analysis and in vitro deubiquitination experiments. V.v.A. and C.v.A. contributed to colony formations assays. I.U. and K.U. generated the RPE-1^CAG129 cell lines and performed repeat expansion measurements under the supervision of G.B. A.H. prepared recombinant USP7 fragments and performed fluorescence polarization assays under the supervision of R.J.H. R.J.H generated AlphaFold3-based structural models and performed model curation and confidence assessment. R.G. performed multiple sequence alignments and conservation analysis of USP7 binding sites in FAN1. A.A.S. and G.C. designed the project with crucial contributions from M.G. and G.B. G.C., G.B., and A.A.S. wrote the manuscript with inputs from all authors.

## Competing interests
