## [Transparent Peer Review file · Nature Communications]

USP7 deubiquitinase stabilizes FAN1 to support DNA crosslink repair and suppress CAG repeat expansion

Corresponding Author: Professor Alessandro Sartori

Version 0:

Reviewer comments:

Reviewer #1

(Remarks to the Author)

The manuscript reports a novel role of USP7 in promoting de-ubiquitination of FAN1.

The authors present a detailed study that shows convincingly the effect of USP7 on FAN1 is direct and specific. They also frame the observation in two biologically relevant contexts ICL repair and CAG repeat expansion.

I only have few minor comments to improve presentation:

1. In Figure 1 E the authors present co-immunoprecipitation of FAN with USP7 using endogenous antibodies. It is not clear what is added in lane 2 of the Western blot. Is this IgG control? Please indicate
2. In the labelling of the Wb the authors indicate GFP or FLAG without clarifying what they aim to detect. The labels indicating the protein expressed in cells are quite small. So, to improve presentation I recommend changing labels as follows: GFP (FAN1) or FAN1 (-GFP) and similar. Same with FLAG.
3. The bipartite recognition by USP7 could be a general mechanism of action, have the authors tried to look at other known substrates of USP7 to detect the bipartite motif?
4. The alphafold modelling adds significance to the proposed interaction model but it could be relevant to add PAE scoring and ipTM scores of the modelled interaction. Have the authors tried to model the bipartite motif in alpha fold ?
5. The discussion feels a bit disjointed between Huntington and cancer. Probably, to maintain cohesion of the manuscript the authors could discuss on the role of repeats expansion in cancer. On the light of the observations presented here, the inhibition of USP7 in p53 null tumours might be highly relevant to control repeat expansions.

Reviewer #2

(Remarks to the Author)

Collotta and colleagues present evidence that stability of the FAN1 nuclease is regulated by the deubiquitylating enzyme USP7 and that loss of USP7 recapitulates many cellular phenotypes previously associated with a lack of FAN1, including the accelerated expansion of CAG repeat sequences. Overall this is a fairly straight forward study, mirroring the experimental approach found in several other studies identifying substrates of USP7. The manuscript is well written. The data presented is of a high quality and supports the conclusions made. I liked the fact that the authors have tried to address the contributions of the other known DDR substrates of USP7 when trying to correlate the phenotypes of USP7 depleted cells to a specific loss of FAN1. My one and only criticism of this manuscript is that data presented in Figure 2, where the authors are trying to map the USP7 binding motifs on FAN1 is essentially a negative result. This is not a criticism of the quality of the data but in panel 2H, it would seem that loss of PXXS and KXXXK motifs of FAN1 makes no difference to its stability. As the authors indicate, this suggests the presence of additional USP7 binding sites. Given that the outcome of the experimentation presented in Figure 2 is negative, I would suggest moving this data to the supplementary data. Furthermore, I think the manuscript would be significantly improved if the authors identify the additional USP7 binding sites, so that a mutant can be made and the phenotype of FAN1 KO cells complemented with a mutant unable to bind USP7 can be documented. This would greatly support that data indicating that the phenotypes tested arising from USP7 loss/knockdown arise specifically as a consequence of FAN1 instability and not destabilisation of another protein.

Reviewer #3

(Remarks to the Author)

In this manuscript, Collota et al. have investigated the regulation of FAN1 protein stability and function in ICL repair and CAG repeat extension in Huntington's disease. Authors have identified USP7 as a new interactor and the deubiquitinase of FAN1 nuclease. Authors demonstrate, USP7-FAN1 interaction in cells and in vitro, wherein USP7 interacts with FAN1 through its N-terminal TRAF and C-terminal UBL domains. USP7 depletion, destabilizes FAN1 protein, leads to reduced FAN1 foci formation and increased cellular sensitivity upon MMC treatment. Further, loss of USP7 accelerates CAG repeat expansion in an HD cellular model. Collectively, authors establish USP7 as a critical regulator of FAN1 activity in genome instability. Identification of USP7 as FAN1 deubiquitinase and regulator is novel and significant. Overall, the study is thorough, experiments are well designed, data supports the conclusions. Authors have provided detailed methodology and manuscript is well written. However, few major and minor queries listed below need to be addressed prior to publication.

1. Figure 3A, comparable levels of USP7 depletion using two different siRNA #1 and #2 is observed in RPE parental and p53KO cells, however, why FAN1 and DNMT1 levels are dramatically decreased in lanes 2, 3 and 6, but not lane 5? Quantification of western blots are missing.
2. Figure 3F and 3G, GFP-FAN1 input blot, why over-expression of USP7 C223S catalytic inactive mutant does not lead to reduction of FAN1 protein levels?
3. Authors show that FAN1 protein levels decreases with USP7 depletion in the absence of MMC treatment (Figure 3A, 3B, and 3D), however, in Figure 4C, FAN1 protein levels appear unchanged upon USP7 depletion (lane 3) compared to control (lane 1) in the absence of MMC treatment. Why? Also, western blot quantification is not shown.
4. FAN1 is recruited to ICLs by ubiquitinated FANCD2 and FANCI. Authors have examined FAN1 and FANCD2 foci formation upon MMC treatment upon USP7 depletion (Figure 4A and B) and conclude that USP7 acts downstream of FANCD2. However, FAN1 foci formation in FANCD2siRNA and FANCD2siRNA + USP7siRNA double knockdown cells have not been shown and compared. Also, does USP7 localize to ICLs, downstream of ubFANCI-FANCD2?
5. Authors show that USP7, but not USP9X, USP11, and USP48, regulates FAN1 protein stability. USP1 is the DUB for ubFANCD2-FANCI. Does USP1 regulate FAN1 ubiquitination and protein stability?
6. Quantification of Western blots in all figures in this manuscript is missing and should be performed.

Version 1:

Reviewer comments:

Reviewer #1

(Remarks to the Author)

The authors have prepared a very extensive review which addresses all my concerns. I commend them on the scientific rigour shown in the revision process.

Reviewer #2

(Remarks to the Author)

The authors have gone to some lengths to identify the USP7 binding sites on FAN1, so despite the fact that they were unable to generate mutant lacking the ability bind USP7, I am satisfied that the authors did their best. Based on this, I would recommend that this manuscript be published.

Reviewer #3

(Remarks to the Author)

Sartori and colleagues have successfully addressed the queries raised during the manuscript review. Identification of USP7 as FAN1 deubiquitinase is significant and the revised study is suitable for publication.

On behalf of all co-authors, we would like to express our sincere gratitude to the reviewers for their thorough evaluation of our manuscript. We appreciate their overall positive feedback and the constructive, thoughtful critiques provided. As detailed below, we have carefully addressed all of the reviewers' comments, and we hope that the revised manuscript now meets the expectations and requirements for publication.

Reviewer #1 (Remarks to the Author):

The manuscript reports a novel role of USP7 in promoting de-ubiquitination of FAN1. The authors present a detailed study that shows convincingly the effect of USP7 on FAN1 is direct and specific. They also frame the observation in two biologically relevant contexts ICL repair and CAG repeat expansion. We thank this reviewer for her/his appreciation of our work.

I only have few minor comments to improve presentation:

1. In Figure 1E the authors present co-immunoprecipitation of FAN with USP7 using endogenous antibodies. It is not clear what is added in lane 2 of the Western blot. Is this IgG control? Please indicate. We thank the reviewer for this question. Indeed, we inadvertently omitted the indication in the figure and its corresponding legend that lane 2 represents the 'beads only' control sample. This has now been corrected in the revised version of the manuscript.

2. In the labelling of the Wb the authors indicate GFP or FLAG without clarifying what they aim to detect. The labels indicating the protein expressed in cells are quite small. So, to improve presentation I recommend changing labels as follows: GFP (FAN1) or FAN1 (α -GFP) and similar. Same with FLAG. This is indeed a very good suggestion. To improve readability, we have now re-labelled all western blots that included tagged proteins as follows: GFP (FAN1), FLAG (USP7) and HA (Ub). Furthermore, we have labelled the corresponding GST-tagged proteins in Figure 1H.

3. The bipartite recognition by USP7 could be a general mechanism of action, have the authors tried to look at other known substrates of USP7 to detect the bipartite motif?

Yes, we tried very hard to fine-map USP7 binding sites in FAN1. As also referee 2 raised a similar concern, I will describe our previous and new efforts here. USP7 primarily engages its N-terminal TRAF-like domain to bind [P/A/E]-X-X-S sequence motifs in target proteins (e.g. p53) (PMID: 28591556). A fewer number of USP7 substrates (e.g. RNF169 and DNMT1) are exclusively recognized by the USP7 UBL1-2 domain for which another consensus binding motif (K-X-X-X-K) has been identified. For only a small subset of USP7 substrates a 'bipartite recognition' mode involving both USP7 binding pockets has been demonstrated, including DNA polymerase iota (POLI) and PAF15 (PMID: 33279577; PMID: 37775071; PMID: 36734974). Interestingly, in the case of PAF15, two TRAF-like and one UBL binding sites had to be mutated at once to abolish USP7 binding, while in the case of POLI, the interaction of the UBL interface was weaker than the TRAF-mediated interaction. Our previous data suggested that FAN1 most likely belongs to the 'bipartite' group, as GFP-FAN1 was co-immunoprecipitated (co-IP) with Flag-tagged constructs expressing either the N-terminal TRAF-like domain or the C-terminal UBL domain (**Figure 2E**). We performed a similar experiment, again using USP7's individual TRAF and UBL domains, and observed that FAN1 interacts equally well with both USP7 fragments, further supporting its bipartite recognition mode that is reminiscent of that of POLI (**Rebuttal Figure 1**).

Rebuttal Figure 1: FAN1 interacts with both the TRAF-like and UBL-domains of USP7. Top panel: Schematic representation of human USP7. TRAF, tumour necrosis factor receptor associated factor; CD, catalytic domain; UBL, ubiquitin-like. Bottom panel: HEK293^{eGFP-FAN1} cells were transfected with either empty vector (e.v.) or the indicated FLAG-USP7 expression constructs. 24 h later, GFP-FAN1 expression was induced with Dox (100 ng/ml). 48 h post induction, cells were lysed and whole-cell extracts were subjected to IP using anti-FLAG M2 affinity resin. Inputs and recovered protein complexes were analysed by immunoblotting. Asterisk indicates IgG heavy chain from anti-FLAG M2 affinity resin.

We next aimed to identify both the TRAF- and the UBL-binding motif(s) in FAN1 following an unbiased systematic approach based on bioinformatics analysis. We were initially focusing on the N-terminal domain (NTD) of FAN1 (aa 1-372) in our search for these motifs, as (i) NTD-deleted FAN1 (aa 373-1017) did not seem to interact with USP7 and (ii) the FAN1-NTD was sufficient to associate with USP7 in cell lysates and *in vitro* (Figures 1G and 1H). Thus, we subjected biotinylated FAN1 peptides containing the most promising [P/A/E]-X-X-S or K-X-X-X-K motifs in the NTD to pulldown assays with recombinant USP7, revealing 181-PQSS-184 and 158-KLSRK-162 as putative TRAF and UBL binding motifs, respectively (Figures 2B and S3B). However, while the S184A 'TRAF3' mutant peptide failed to bind USP7 (Figure 2C), our new findings reveal that the K158A/K162A 'UBL2' mutant peptide exhibited no reduction in USP7 binding (Rebuttal Figure 2; new Figure S3C). These results suggests that 158-KLSRK-162 sequence is unlikely to represent a *bona fide* UBL-binding motif.

Rebuttal Figure 2 (new Figure S3C): FAN1-KLSRK sequence is not a true UBL binding motif. Biotinylated FAN1 UBL2-wt or -K158A/K162A (2KA) 13mer peptides (10 μg) were immobilized on Strep-Tactin[®] XT 4Flow[®] resin and incubated with purified recombinant His-USP7. Inputs and recovered His-USP7 were analysed by immunoblotting.

In the revised manuscript, we have incorporated new fluorescence polarisation (FP) data that provide strong support for our conclusion that the 181-PQSS-184 sequence in FAN1 constitutes a *bona fide* PxxS TRAF-binding motif. As shown in Rebuttal Figure 3 (new Figure 2D), an 8mer peptide containing this motif binds efficiently to the USP7 TRAF domain, whereas the S184A mutant displays markedly

reduced binding. Notably, the P181A mutation did not abolish USP7 binding, consistent with the fact that an AxxS sequence still conforms to the TRAF consensus sequence.

Rebuttal Figure 3 (new Figure 2D): FAN1 contains a consensus TRAF-binding motif (181-PQSS-184). Fluorescence polarization binding curves of indicated FITC-labelled FAN1-derived 8mer peptides measured as a function of USP7-TRAF (aa 2–207) concentration. Fluorescence polarization values were normalized by expressing the millipolarization (mP) of each sample relative to the corresponding wild-type peptide. Background signal was corrected by subtracting baseline FP measured in the absence of USP7-TRAF. Data represent mean \pm SD (n = 3).

In our previous mutagenesis experiments, deletion of residues 157-207 within the FAN1-NTD fragment (aa 1-290) substantially reduced USP7 binding (**Figure S3D**). However, the same 50-amino acid deletion in the context of full-length FAN1 did not affect the USP7 interaction (**Figure S3E**). These data indicate that USP7 can associate with FAN1 independently of the 158-KLSRK-162 and 181-PQSS-184 motifs. We also observed that the FAN1-TRAF4 peptide containing the 358-EQGS-361 motif shows weak USP7 binding (**Figures 2B and S2B**). Consistently, AlphaFold3 (AF3) predicts a plausible interface between the USP7 TRAF-like domain and the FAN1-TRAF4 segment (aa 355-366), with ipTM and pTM scores only slightly lower than those of the FAN1-TRAF3 peptide (aa 179-186) (**Rebuttal Figure 4A and Figure S2C**). Preliminary FP assays using FITC-labelled FAN1-derived 12mer peptides further suggest that FAN1 358-EQGS-362 may constitute an alternative TRAF-binding site (**Rebuttal Figure 4B**). Nevertheless, full-length FAN1 carrying mutations in both putative TRAF interaction motifs (S184A/S361A/S362A) retained robust interaction with USP7 (**Rebuttal Figure 4C**). Together with our *in vitro* crosslink-mass spectrometry analysis (see response to Ref. 2), these data support the notion that FAN1 contains multiple USP7-contact regions beyond those present in the NTD. However, because these findings are preliminary and do not alter our main conclusions, we have not included them in the revised manuscript.

Rebuttal Figure 4: FAN1 may contain a second, alternative TRAF-binding motif (358-EQGS-361) in the NTD. (A) Schematic representation of the 3D structure of the interaction complex between USP7's TRAF domain (aa 68-195) and FAN1 TRAF4 peptide (aa 355-366) by AlphaFold 3, coloured according to the predicted local Distance Difference Test (pIDDT). Dark blue: regions with pIDDT greater than 90 (Very high); light blue: regions with pIDDT between 70 and 90 (Confident); yellow regions with pIDDT between 50 and 70 (Low); orange: regions with pIDDT lower than 50 (Very low). Interface predicted template modelling (ipTM) = 0.91 and predicted template modelling (pTM) = 0.92. Predicted Aligned Error (PAE) plot indicating regions of high (dark green) and low (light green) confidence. (B) Fluorescence polarization binding curves of the indicated FITC-labelled FAN1-derived 12mer peptides measured as a function of USP7-TRAF (aa 2–207) concentration. Fluorescence polarization values were normalized by expressing the millipolarization (mP) of each sample relative to the corresponding wild-type peptide. Background signal was corrected by subtracting baseline FP measured in the absence of USP7-TRAF (0 μM). Data represent mean \pm SD (n = 3). (C) HEK293 cells were transfected with either empty vector (e.v.), GFP-FAN1-wt or GFP-FAN1 3SA (S184A/S361A/S362A) expression constructs. 48 h after transfection, cells were lysed and whole-cell extracts were subjected to GFP-Trap resin. Inputs and recovered protein complexes were analysed by immunoblotting.

4. The alphafold modelling adds significance to the proposed interaction model but it could be relevant to add PAE scoring and ipTM scores of the modelled interaction. Have the authors tried to model the bipartite motif in alpha fold?

We do agree with the reviewer that confidence metrics of AlphaFold (AF) predictions are essential for interpreting the models. Accordingly, we indicated color-coded predicted local Distance Difference Test (pIDDT) values, together with the ipTM and pTM scores (Figure S2C, middle). In addition, we included a Predicted Aligned Error (PAE) plot indicating regions of high (dark green) and low (light green) confidence (Figure S2C, right). For improved clarity, the corresponding confidence values are now explicitly included in the corresponding figure legend.

When modelling putative interactions between the USP7 UBL domain and candidate FAN1 sequence binding motifs using AF3, none of predicted complexes reached high-confidence levels. Likewise, AF3 modelling of full-length FAN1 in complex with USP7 did not yield confident structural predictions. These results are consistent with publicly available predictions (<https://predictomes.org/summary/qk29jdc>), where two of three FAN1-USP7 models meet minimum confidence thresholds (PAE < 15; pIDDT > 50) but fall short of high-confidence criteria (SPOC < 0.75). Among these models, FAN1 residue D61 was predicted to have the strongest potential for forming hydrogen bonds or salt bridges with USP7 residue R788 in the UBL domain. To experimentally test this

possibility, we generated a FAN1-D61R mutant; however, this mutation did not reduce FAN1-USP7 binding (**Rebuttal Figure 5**).

Rebuttal Figure 5: HEK293T cells were transfected with either empty vector (e.v.) or the indicated GFP-FAN1 expression constructs. 48 h after transfection, cells were lysed and whole-cell extracts were subjected to GFP-Trap resin. Inputs and recovered protein complexes were analysed by immunoblotting.

As outlined in detail before, we were successful in identifying a FAN1 sequence motif recognized by the TRAF-like substrate recognition domain of USP7, while FAN1 site(s) interacting via the UBL domain of USP7 have remained elusive.

5. The discussion feels a bit disjointed between Huntington and cancer. Probably, to maintain cohesion of the manuscript the authors could discuss on the role of repeats expansion in cancer. On the light of the observations presented here, the inhibition of USP7 in p53 null tumours might be highly relevant to control repeat expansions.

We are grateful for this important suggestion. Although Huntington's disease (HD), other repeat expansion disorders (REDs) and cancer are clinically distinct conditions, emerging evidence indicates that the length of certain trinucleotide repeat tracts can inversely correlate with cancer risk. For example, shorter CAG and GGN repeats in the androgen receptor gene have been associated with an increased risk of prostate cancer (PMID: 28091563). Likewise, several studies report an inverse correlation between the CAG repeat length in exon 1 of *HTT* (huntingtin) and overall cancer incidence in HD patients (PMID: 22503213, PMID: 28202696, PMID: 10506723, PMID: 23017147), with only two studies suggesting a potential tumour-promoting effect of mutant huntingtin (*mHTT*) in breast or melanoma models (PMID: 23300147, PMID: 28202696). Based on this, one could speculate that USP7 inhibition might theoretically lower cancer risk by promoting CAG repeat expansion. However, repeat expansion proceeds slowly over many years, making it unlikely that USP7 inhibition could be deployed as a practical cancer-prevention strategy, as it would require long-term treatment to reach a repeat length conferring measurable protection.

By contrast, USP7 inhibition has been extensively explored as an anticancer approach through more immediate mechanisms, most notably its regulation of the MDM2/MDMX-p53 axis. By promoting MDM2 degradation, USP7 inhibitors activate p53, and, consequently, p21, and can induce apoptosis in p53-proficient cancer cells (PMID: 36186590). In addition, USP7 controls tumour growth via p53-independent pathways involving substrates such as PTEN, N-Myc, FOXP3, FOXM1 and KRAS (PMID: 18716620, PMID: 27618649, PMID: 30697058, PMID: 37291217, PMID: 39499616).

Our work highlights FAN1 as an additional USP7 target protein, directly linking USP7-dependent deubiquitination to both interstrand crosslink (ICL) repair and repeat instability. In p53-deficient tumours, USP7 inhibition therefore remains therapeutically attractive: by destabilising FAN1, it can enhance sensitivity to ICL-inducing agents or replication stress. In post-mitotic tissues such as the brain, however, reduced FAN1 stability has the opposite consequence, exacerbating CAG repeat expansion as we observed in our cellular CAG repeat expansion models. This reveals a context-dependent trade-off: the same intervention that enhances genotoxic sensitivity in cancer can compromise repeat stability in neurons.

Taken together, although USP7 inhibition may influence repeat expansion dynamics, we consider it unlikely to be a viable strategy for cancer prevention, given the slow kinetics of repeat lengthening and the risk of aggravating repeat expansion disorders. Instead, current data support USP7 inhibition

primarily as an anticancer therapy in settings where tumours depend on its substrates, while strategies aimed at stabilising FAN1, potentially via more selective USP7 modulation or substrate-specific targeting, may offer a complementary route to mitigate repeat-driven neurodegeneration.

Following the reviewer's suggestion to improve the cohesion of our discussion, we have added the following paragraph on page 14:

“Intriguingly, emerging evidence suggests that trinucleotide repeat length may influence cancer susceptibility, with longer repeats, such as the expanded CAG tract in exon 1 of mutant huntingtin (mHTT), associated with reduced cancer incidence in HD cohorts^{74,75}. Our findings place FAN1 at a mechanistic crossroads: USP7-dependent FAN1 stabilisation maintains genome integrity in proliferating cells while limiting repeat expansion in post-mitotic neurons. Consequently, USP7 inhibition could have context-dependent effects, enhancing chemosensitivity in p53-deficient tumours, yet potentially accelerating TNR instability in the nervous system, highlighting the USP7-FAN1 axis as a molecular link between cancer biology and repeat expansion disorders.”

Reviewer #2 (Remarks to the Author):

Collotta and colleagues present evidence that stability of the FAN1 nuclease is regulated by the deubiquitylating enzyme USP7 and that loss of USP7 recapitulates many cellular phenotypes previously associated with a lack of FAN1, including the accelerated expansion of CAG repeat sequences. Overall, this is a fairly straight forward study, mirroring the experimental approach found in several other studies identifying substrates of USP7. The manuscript is well written. The data presented is of a high quality and supports the conclusions made. I liked the fact that the authors have tried to address the contributions of the other known DDR substrates of USP7 when trying to correlate the phenotypes of USP7 depleted cells to a specific loss of FAN1.

We thank this reviewer for the positive evaluation of our work.

My one and only criticism of this manuscript is that data presented in Figure 2, where the authors are trying to map the USP7 binding motifs on FAN1 is essentially a negative result. This is not a criticism of the quality of the data but in panel 2H, it would seem that loss of PXXS and KXXXK motifs of FAN1 makes no difference to its stability. As the authors indicate, this suggests the presence of additional USP7 binding sites. Given that the outcome of the experimentation presented in Figure 2 is negative, I would suggest moving this data to the supplementary data.

We agree with the reviewer that the data presented in the original Figure 2 is largely negative, particularly regarding our GFP-trap pulldown experiments with putative USP7-binding deficient FAN1 mutants. Accordingly, we have moved the corresponding panels to the Supplementary information (**Figures S3A, D and E**). Similarly, as our new streptavidin pulldown data using a mutant FAN1-UBL2 biotinylated peptide indicate that the 158-KLSRK-162 sequence is unlikely to constitute a 'bona fide' UBL binding motif (see **Rebuttal Figure 2**, addressing #3 of Ref.1), we have moved also these data to the Supplementary information (**Figures S3B and C**). In contrast, our new fluorescence polarisation (FP) data firmly demonstrate that the 181-PQSS-184 tetrapeptide in FAN1 is as a *bona fide* TRAF-binding motif (see **Rebuttal Figure 3; new Figure 2D**, addressing #3 of Ref.1). Therefore, we have retained the corresponding 'positive' data on the USP7(TRAF)-FAN1 interaction, including a Flag-USP7 IP experiment showing that the USP7 UBL domain binds to FAN1, in the revised manuscript (**Figure 2E**).

Furthermore, I think the manuscript would be significantly improved if the authors identify the additional USP7 binding sites, so that a mutant can be made and the phenotype of FAN1 KO cells complemented with a mutant unable to bind USP7 can be documented. This would greatly support that data indicating that the phenotypes tested arising from USP7 loss/knockdown arise specifically as a consequence of FAN1 instability and not destabilisation of another protein.

We agree that demonstrating separation-of-function phenotypes using FAN1 KO cells complemented with a USP7-binding deficient FAN1 mutant would provide additional mechanistic insight. We also hope the reviewer recognizes our extensive efforts to identify both USP7 TRAF-like and UBL-binding sites in FAN1 (see **Rebuttal Figures 4 and 5**, addressing #3 of Ref.1), which collectively support the presence of multiple FAN1-USP7 interaction interfaces.

To further investigate the molecular basis of the FAN1-USP7 interaction, we performed unbiased cross-linking mass spectrometry (XL-MS) using purified recombinant full-length FAN1 and USP7. Proteins were cross-linked with Bis(sulfosuccinimidyl)Suberate (BS3) (**Rebuttal Figure 6**), digested with trypsin, and analysed by LC-MS/MS on a calibrated Fusion Lumos mass spectrometer. Data were searched against the corresponding protein sequences using the Spectrum Identification Machine (SIM1). The experiment was performed twice independently, and cross-linked peptides identified in both replicates with a spectral count >1 are listed in the **Rebuttal Table 1**.

Rebuttal Figure 6: SDS-PAGE analysis of crosslinked FAN1-USP7 complexes. One microgram each of full-length recombinant FAN1 purified from Sf9 insect cells (lane 1) and USP7 purified from bacteria (lane 2) was incubated with increasing amounts of BS3 (lanes 3-6) for 30 min at room temperature. The reaction was quenched by addition of Tris HCl pH 7.5 at a final concentration of 47 mM. Based on this titration, a two-fold molar excess of BS3 (lane 4) was selected as the condition for the XL-MS analysis.

USP7		FAN1		Exp.1	Exp.2
Peptide sequence 1	aa	Peptide sequence 2	aa	spectra	spectra
AGEQQLSEPEDMEMEAGDTDDPPR	12-35	ENVFKCDSLKEECIP	213-227	8	2
DPANYILHAVLVHSGDNHGGHYVVY LNPKGDGKWCK	444-479	LRESPCK	753-760	8	4
IQDYDVSLDKALDELMDGDIIVFQK	746-770	ELESLLSQRIYCPDSR	689-704	10	4

Rebuttal Table 1: List of the crosslinks identified between USP7 and FAN1. Peptide sequences pairs identified in two independent XL-MS analyses are listed with the residue position and spectral counts.

Our preliminary XL-MS analysis revealed three major crosslinks, between (1) a region proximal to the TRAF-like domain of USP7 (aa 12-35) and the FAN1-NTD (aa 213-227), (2) the catalytic domain (CD) of USP7 (aa 444-479) and the FAN1-TPR domain (aa 753-760), and (3) the Ubl2 domain of USP7 and the FAN1-TPR domain (aa 689-704). Notably, the cross-linked peptide of the USP7-CD included two invariant histidine residues, H456 and H464, required for deubiquitination activity (PMID: 12507430). Although the XL-MS data did not pinpoint any canonical TRAF- or UBL-domain sequence binding motifs in FAN1, pull-down experiments with recombinant proteins confirmed that all three major USP7 domains are capable of interacting with FAN1 *in vitro*, with the TRAF domain showing the strongest binding (**Rebuttal Figure 7**).

Rebuttal Figure 7: All three major domains of USP7 interact with FAN1. Purified recombinant biotin-tagged USP7 fragments (6 µg) were immobilized on Strep-Tactin®XT-4Flow® resin and incubated with purified recombinant MBP-His-tagged FAN1 (1 µg). Ponceau staining was performed to visualize the USP7 fragments. Inputs and recovered MBP-His-FAN1 were analysed by immunoblotting. Arrowheads indicate the protein band corresponding to each USP7 fragment.

As a final cautionary note, even if we had successfully identified a FAN1 mutant that no longer interacts with USP7 and consequently exhibits reduced stability, we would not expect substantial differences in phenotypes between *FAN1*^{-/-} KO cells complemented with wild-type FAN1 or this mutant. This is because, in our doxycycline-inducible system, both variants would be expressed at levels far exceeding those of endogenous FAN1, thereby effectively rescuing the ICL repair and CAG repeat expansion defects observed in FAN1-depleted cells. Moreover, overexpression of FAN1, an active endo- and exonuclease, might itself be cytotoxic. Consistent with this, we previously reported that MMC sensitivity of cells (over)expressing an MLH1-binding deficient FAN1 mutant was partially rescued when the mutant was rendered nuclease-defective (PMID: 34330701). For these reasons, we believe that the impact of studying a USP7-binding-defective FAN1 mutant can only be meaningfully assessed when the mutation is introduced at the endogenous locus. Due to time constraints and the lack of well-defined candidate residues for precise genome editing, this approach is not feasible within the timeframe of the current study.

Reviewer #3 (Remarks to the Author):

In this manuscript, Collota et al. have investigated the regulation of FAN1 protein stability and function in ICL repair and CAG repeat extension in Huntington's disease. Authors have identified USP7 as a new interactor and the deubiquitinase of FAN1 nuclease. Authors demonstrate, USP7-FAN1 interaction in cells and in vitro, wherein USP7 interacts with FAN1 through its N-terminal TRAF and C-terminal UBL domains. USP7 depletion, destabilizes FAN1 protein, leads to reduced FAN1 foci formation and increased cellular sensitivity upon MMC treatment. Further, loss of USP7 accelerates CAG repeat expansion in an HD cellular model. Collectively, authors establish USP7 as a critical regulator of FAN1 activity in genome instability. Identification of USP7 as FAN1 deubiquitinase and regulator is novel and significant. Overall, the study is thorough, experiments are well designed, data supports the conclusions. Authors have provided detailed methodology and manuscript is well written. However, few major and minor queries listed below need to be addressed prior to publication.

We thank the reviewer for her/his positive assessment of our work.

1. Figure 3A, comparable levels of USP7 depletion using two different siRNA #1 and #2 is observed in RPE parental and p53KO cells, however, why FAN1 and DNMT1 levels are dramatically decreased in lanes 2, 3 and 6, but not lane 5? Quantification of western blots are missing.

We thank the reviewer for raising these important points. We have now included quantification of the relevant proteins in all western blots. In Figure 3A, we consistently observed that siRNA #2 targeting USP7 caused a stronger reduction in FAN1 (and DNMT1) levels compared to siRNA #1. Moreover, both FAN1 and DNMT1 showed a more pronounced decrease in the RPE-1 parental cell line than in the p53 KO cells. We attribute this difference primarily to a slightly lower USP7 knockdown efficiency in the p53 KO line. Additionally, the stronger effect observed in the parental line may partly result from the marked G1 cell-cycle arrest caused by USP7 knock-down (Figure S4A). Importantly, in USP7-depleted RPE-1 cells, reduced FAN1 levels were restored by proteasomal inhibition, further supporting USP7's role in preventing proteasome-mediated degradation of FAN1 (Rebuttal Figure 8; new Figure S4C).

Rebuttal Figure 8 (and new Figure S4C): USP7 protects FAN1 from proteasomal degradation in RPE-1 cells. RPE-1 parental cells were transfected with either non-targeting (Ctrl) or USP7 siRNA oligos. 48 h later, cells were either mock-treated (-) or treated (+) with MG-132 (10 μ M) for 6 h and whole-cell lysates were analysed by immunoblotting. Lanes 3 and 4 were rearranged, as they were not adjacent to lane 1 and 2 in the original blot. Relative FAN1 protein levels were determined by quantification of FAN1 band intensity (normalized to α -Tubulin) with the ImageJ software.

2. Figure 3F and 3G, GFP-FAN1 input blot, why over-expression of USP7 C223S catalytic inactive mutant does not lead to reduction of FAN1 protein levels?

This result is indeed a bit puzzling. Based on the results obtained in U2OS cells shown in the original Figure 3E (Rebuttal Figure 9A), one would indeed expect that overexpression of the catalytic-dead USP7 mutant would similarly reduce GFP-FAN1 levels in HEK293 (Figures 3F and G). To address this discrepancy, we repeated the experiment in U2OS cells two additional times and found that overexpression of wild-type Flag-USP7 consistently stabilizes GFP-FAN1, whereas overexpression of the Flag-C223S mutant produces a more variable phenotype, likely depending on its ability to exert a dominant-negative effect (Rebuttal Figures 9B-D). We also noted that USP7 C223S was expressed at markedly lower levels than wild-type USP7, although still clearly above endogenous USP7. To better

illustrate these two observations, we have replaced the original Figure 3E with the western blot now shown in **Rebuttal Figure 9C**. Although the USP7 C223S mutant has been reported to exert dominant-negative effects in certain contexts (PMID: 32884836), several studies have also shown that its expression does not affect the stability of other USP7 substrates (PMID: 30367141). As we do not observe any change in GFP-FAN1 levels in stable HEK293 cells co-transfected with HA-tagged ubiquitin, irrespective of which USP7 variant is overexpressed (**Figures 3F and G**), we believe that this system is well suited for dissecting ubiquitination of substrates but may not be optimal for assessing protein stability. Although HEK293 cells offer high transfection efficiency, the simultaneous overexpression of three proteins (GFP-FAN1, Flag-USP7 and HA-Ub) can create non-physiological conditions, including saturation or overload of ubiquitin-dependent pathways or altered stoichiometry among the proteins of interest. Such effects could mask potential changes in FAN1 stability.

Rebuttal Figure 9 (panel C is replacing Figure 3E): Overexpression of catalytically-active USP7 stabilizes FAN1. (A-C) Doxycycline-inducible U2OS^{GFP-FAN1} cells were transfected with either empty vector (e.v.) or indicated FLAG-USP7 expression constructs. 24 h later, GFP-FAN1 expression was induced with Dox (100 ng/ml). 24 h post induction, whole-cell lysates were analysed by immunoblotting. Relative GFP-FAN1 protein levels were determined by quantification of GFP-FAN1 band intensity (GAPDH) with the ImageJ software. (D) Bar-chart indicating relative GFP-FAN1 levels from the quantifications in A-C. Dots indicate single values, bars represent mean \pm SD.

3. Authors show that FAN1 protein levels decreases with USP7 depletion in the absence of MMC treatment (Figure 3A, 3B, and 3D), however, in Figure 4C, FAN1 protein levels appear unchanged upon USP7 depletion (lane 3) compared to control (lane 1) in the absence of MMC treatment. Why? Also, western blot quantification is not shown.

We thank the reviewer for raising this point. We have now included quantification of FAN1 protein levels in Figure 4C, which allows appreciation of the mild reduction in FAN1 levels following USP7 depletion in the absence of MMC treatment. To further address this concern and rule out contributions from residual USP7 after short-term siRNA transfection, we initiated the generation of USP7^{-/-} KO HeLa cell lines using CRISPR during the revision process. In two independent USP7 KO clones, we observed a pronounced decrease in FAN1 protein abundance in the absence of MMC treatment (**Rebuttal Figure 10**). These results indicate that, in HeLa cells, FAN1 stability is more strictly regulated by USP7, and substantial FAN1 destabilization in the absence of ICL damage occurs only upon complete USP7 loss.

Rebuttal Figure 10: USP7 stabilizes FAN1 in HeLa cells. Western blot analysis of lysates derived from HeLa parental and two USP7 KO clones. Relative FAN1 protein levels were determined by quantification of FAN1 band intensity (normalized to GAPDH) with the ImageJ software.

4. FAN1 is recruited to ICLs by ubiquitinated FANCD2 and FANCI. Authors have examined FAN1 and FANCD2 foci formation upon MMC treatment upon USP7 depletion (Figure 4A and B) and conclude that USP7 acts downstream of FANCD2. However, FAN1 foci formation in FANCD2siRNA and FANCD2siRNA + USP7siRNA double knockdown cells have not been shown and compared. Also, does USP7 localize to ICLs, downstream of ubFANCI-FANCD2?

We thank the reviewer for raising these two important points. To directly compare the effects of FANCD2 and USP7 depletion on FAN1 foci formation, we have now performed immunofluorescence (IF) microscopy analysis in U2OS^{GFP-FAN1} cells, either mock-treated or exposed to 150 ng/ml MMC for 24 h. Following the reviewer's suggestion, we included siRNAs targeting FANCD2, USP7, or both in combination. GFP-FAN1 nuclear foci were analysed and quantified via Quantitative Image-Based Cytometry (QIBC). As shown in **Rebuttal Figure 11**, depletion of either USP7 or FANCD2 markedly reduced FAN1 foci following MMC treatment, with FANCD2 knockdown producing a slightly stronger effect, consistent with its established role in FAN1 recruitment to ICLs. Moreover, co-depletion of USP7 and FANCD2 nearly completely abolished MMC-induced FAN1 foci formation, suggesting that USP7 and FANCD2 regulate FAN1 local accumulation at DNA damage through distinct mechanisms. We consider these new findings sufficiently relevant to include them in the revised manuscript (**new Figure 4A**). Accordingly, the original Figure 4A, showing the impact of two different USP7 siRNAs on FAN1 foci, has been moved to the Supplementary information (**Figure S5A**).

Rebuttal Figure 11 (and new Figure 4A): USP7 contributes to efficient FAN1 accumulation at sites of crosslink-induced damage. QIBC analysis of chromatin-bound GFP-FAN1 foci in U2OS^{GFP-FAN1} cells transfected with either non-targeting (Ctrl), USP7 or FANCD2 siRNA oligos. Cells were either mock-treated (-) or treated (+) with MMC (150 ng/ml) for 24 h. Color-coded scatterplots indicate the number of GFP-FAN1 foci per nucleus. Mean (solid line) and standard deviation (SD) from the mean (dashed lines) are indicated. Representative images are shown below. Scale bar, 10 μm.

To address the second point concerning whether USP7 localizes to ICLs, and, if so, whether this occurs downstream of the ubFANCI-FANCD2 complex (ubID2), we performed IF experiments in parental U2OS cells treated or not with 150 ng/ml MMC for 24 h. QIBC analysis revealed that USP7 forms nuclear foci, which increased modestly in numbers upon MMC treatment and were largely unaffected by FANCD2 depletion (**Rebuttal Figure 12**). These data suggest that USP7 localizes to MMC-induced DNA damage sites (or stalled replication forks) independently of the ubID2 complex. Consistent with this, USP7 has previously been shown to be recruited to DSBs upon high-dose X-ray irradiation in an MRN-MDC1-dependent manner (PMID: 30179224) and to regulate multiple DNA repair pathways, some of which contribute to the coordination of ICL repair (PMID: 32850836). Interestingly, USP7 interacts with and stabilizes UHRF1, an E3 ubiquitin ligase and key epigenetic regulator (PMID: 21745816, PMID: 22411829), which has also been reported to act as a sensor for ICLs and to facilitate

FANCD2 recruitment (PMID: 25801034). Therefore, in agreement with our data, it is tempting to speculate that UHRF1 may recruit USP7 to ICLs independently of the ubiD2 complex.

Rebuttal Figure 12: USP7 forms nuclear foci independently of FANCD2. QIBC analysis of endogenous USP7 foci in U2OS cells transfected with either non-targeting (siCtrl) or the indicated siRNA oligos. Cells were either mock-treated (-) or treated (+) with MMC (150 ng/ml) for 24 h. Color-coded scatterplots indicate the number of USP7 foci per nucleus. Mean (solid line) and standard deviation (SD) from the mean (dashed lines) are indicated.

5. Authors show that USP7, but not USP9X, USP11, and USP48, regulates FAN1 protein stability. USP1 is the DUB for ubFANCD2-FANCI. Does USP1 regulate FAN1 ubiquitination and protein stability?

This is indeed a valid point. We initially chose not to include USP1 in our study because it was not a significant hit in our AP-MS screen for FAN1 interactors. However, we have now performed GFP-Trap pulldowns in HEK293^{eGFP-FAN1} cells and found that USP1 can indeed interact with FAN1, although the association appears slightly less prominent than with USP7 (**Rebuttal Figure 13A**). Thus, given its well-established role as a key regulator of DNA repair pathways, including its function in deubiquitinating monoubiquitinated FANCD2 and PCNA during ICL repair and translesion synthesis (PMID: 39848025), we sought to evaluate whether USP1 also regulates FAN1 ubiquitination and protein stability. In marked contrast to USP7, however, USP1 depletion or inhibition only modestly decreased GFP-FAN1 protein levels (**Rebuttal Figures 13B and C**). To formally exclude a prominent role of USP1 in FAN1 deubiquitination, we overexpressed FLAG-USP7 or FLAG-USP1 in HEK293^{eGFP-FAN1} cells co-transfected with HA-Ub. We found that only overexpression of USP7 significantly reduced FAN1 polyubiquitination, whereas USP1 had no effect (**Rebuttal Figure 13D**). Taken together, these preliminary data suggest that USP1 may influence FAN1 function in ICL repair but does not so by regulating FAN1 protein stability. Therefore, unless the reviewer raises a compelling reason, we propose not to include this data in the Supplementary information of the revised manuscript.

Rebuttal Figure 13: USP1 interacts with FAN1 but does not modulate FAN1 stability and ubiquitination. (A) HEK293^{eGFP-FAN1} cells were induced with doxycycline (Dox, 100 ng/ml). 48 h later, cells were lysed and whole-cell extracts were subjected to GFP-Trap resin. Inputs and recovered protein complexes were analysed by immunoblotting. (B) Doxycycline-inducible U2OS^{eGFP-FAN1} cells were transfected with either non-targeting (Ctrl) or USP1 siRNA oligos. 24 h later, GFP-FAN1 expression was induced with Dox (100 ng/ml) for 24 h and whole-cell lysates were analysed by immunoblotting. Relative FAN1 protein levels were determined by quantification of FAN1 band intensity (normalized to GAPDH) with the ImageJ software. (C) Same cells as in A were induced with Dox (100 ng/ml). 24 h later, cells were either mock-treated with DMSO (-) or treated (+) with 10 μ M of either GNE-6640 (USP7i) or ML323 (USP1i) for 24 h and whole-cell lysates were analysed by immunoblotting. Relative FAN1 protein levels were determined by quantification of FAN1 band intensity (normalized to GAPDH) with the ImageJ software. (D) Doxycycline-inducible HEK293^{eGFP-FAN1} cells were co-transfected with either empty vector (e.v.) or HA-Ubiquitin wt and either e.v., FLAG-USP7 or FLAG-USP1 expression constructs. 24 h later, GFP-FAN1 expression was induced with Dox (100 ng/ml). 24 h post induction, cells were lysed and whole-cell extracts were subjected to GFP-Trap resin. Inputs and recovered protein complexes were analysed by immunoblotting.

6. Quantification of Western blots in all figures in this manuscript is missing and should be performed. We fully agree with the reviewer and have now included quantification of all relevant western blots in the revised manuscript figures.

Author note: During our internal review, we identified and corrected misnumbered tick values in Figure 5B ("Days in culture"), which were shown as -10 -20 -30 instead of the correct -20 -40 -60.

We sincerely thank all three reviewers for their positive feedback on our revised manuscript.
There were no additional requests from the reviewers to be addressed in a point-by-point response.